# Contextual Bandits and Imitation Learning
# with Preference-Based Active Queries

**Ayush Sekhari** [*] [1]  **Karthik Sridharan** [2]  **Wen Sun** [2]  **Runzhe Wu** [2]

## Abstract

We consider the problem of contextual bandits and imitation learning, where the learner lacks direct knowledge of the executed action's reward. Instead, the learner can actively request the expert at each round to compare two actions and receive noisy preference feedback. The learner's objective is two-fold: to minimize regret associated with the executed actions, while simultaneously, minimizing the number of comparison queries made to the expert. In this paper, we assume that the learner has access to a function class that can represent the expert's preference model under appropriate link functions and present an algorithm that leverages an online regression oracle with respect to this function class. For the contextual bandit setting, our algorithm achieves a regret bound that combines the best of both worlds, scaling as $O(\min\{\sqrt{T}, d/\Delta\})$, where $T$ represents the number of interactions, $d$ represents the eluder dimension of the function class, and $\Delta$ represents the minimum preference of the optimal action over any suboptimal action under all contexts. Our algorithm does not require the knowledge of $\Delta$, and the obtained regret bound is comparable to what can be achieved in the standard contextual bandits setting where the learner observes reward signals at each round. Additionally, our algorithm makes only $O(\min\{T, d^2/\Delta^2\})$ queries to the expert. We then extend our algorithm to the imitation learning setting, where the agent engages with an unknown environment in episodes of length $H$, and provide similar guarantees regarding regret and query complexity. Interestingly, with preference-based feedback, our imitation learning algorithm can learn a policy outperforming

a sub-optimal expert, matching the result from interactive imitation learning algorithms (Ross & Bagnell, 2014) that require access to the expert's actions and also reward signals.

## 1. Introduction

Human feedback for training machine learning models has been widely used in many scenarios, including robotics (Ross et al., 2011; 2013; Jain et al., 2015; Laskey et al., 2016; Christiano et al., 2017) and natural language processing (Stiennon et al., 2020; Ouyang et al., 2022). By integrating human feedback into the training process, these techniques align machine learning models with human intention and enable high-quality human-machine interaction (e.g., ChatGPT).

Existing methods generally leverage two types of human feedback. The first is the action from human experts, which is the dominant feedback mode used in the literature of imitation learning or learning from demonstrations (Abbeel & Ng, 2004; Ziebart et al., 2008; Daumé et al., 2009; Ross et al., 2011; Ross & Bagnell, 2014; Sun et al., 2017; Osa et al., 2018; Li et al., 2023). The second type of feedback, preference-based feedback, involves comparing pairs of actions. In this approach, the expert provides feedback by indicating their preference between two options selected by the learner. While both types of feedback have their applications, our focus in this work is on preference-based feedback, which is particularly suitable for scenarios where it is challenging for human experts to recommend the exact optimal action while making pairwise comparisons is much easier.

Learning via preference-based feedback has been extensively studied, particularly in the field of *dueling bandits* (Yue & Joachims, 2011; Yue et al., 2012; Zoghi et al., 2014; Ailon et al., 2014; Komiyama et al., 2015; Wu & Liu, 2016; Saha & Gaillard, 2021; Bengs et al., 2021; Saha & Gaillard, 2022) and *contextual dueling bandits* (Dudík et al., 2015; Saha & Krishnamurthy, 2022; Wu et al., 2023). Different from the standard bandit setting, the learner proposes two actions in dueling bandits and only gets noisy preference feedback from the human expert. Follow-up works extend

[*]Authors are sorted alphabetically by the last name [1]Massachusetts Institute of Technology [2]Cornell University. Correspondence to: Runzhe Wu <rw646@cornell.edu>.

*Interactive Learning with Implicit Human Feedback Workshop at the $40^{th}$ International Conference on Machine Learning*, Honolulu, Hawaii, USA. PMLR 202, 2023. Copyright 2023 by the author(s).

the preference-based learning model from the one-step bandit setting to the multi-step decision-making (e.g., IL and RL) setting (Chu & Ghahramani, 2005; Sadigh et al., 2017; Christiano et al., 2017; Lee et al., 2021a; Chen et al., 2022; Saha et al., 2023). These studies mainly focus on how to learn a high-quality policy from human feedback, without concerning the question of active query in order to minimize the query complexity.

However, query complexity is an important metric to optimize when learning from human feedback, as human feedback is expensive to collect. For instance, InstructGPT (Ouyang et al., 2022) is trained only on around 30K pieces of human feedback, which is significantly fewer than the internet-scale dataset used for pre-training the base model GPT3, indicating the challenge of scaling up the size of human feedback datasets. In other areas, such as robotics, learning from human feedback is also not easy, and prior studies (e.g., Cohn et al. (2011); Zhang et al. (2022); Myers et al. (2023)) have explored this issue from various perspectives. Ross et al. (2013); Laskey et al. (2016) pointed out that querying human feedback in the learning loop is challenging, and extensively querying for feedback puts too much burden on the human experts.

In this work, we design *principled algorithms that can learn from preference-based feedback while at the same time minimizing query complexity*, under the settings of contextual bandits (Auer et al., 2002; Langford & Zhang, 2007) and imitation learning (Ross et al., 2011). Our **main contributions** can be summarized as follows.

- In the contextual dueling bandits setting, the stochastic preference feedback is generated based on some preference matrix (Saha & Krishnamurthy, 2022). We propose an algorithm (named AURORA – in short of Active preference qUeRy fOR contextual bAndits) that can achieve a best-of-both-worlds regret bound (i.e., achieves the minimum of the worst-case regret and an instance dependent regret), while at the same providing an instance-dependent query complexity bound. For benign instances, our regret and query complexity bounds both scale with $\ln(T)$ where $T$ is the total number of interactions in contextual bandits.

- In imitation learning, the stochastic preference feedback is generated based on the underlying reward-to-go of the expert's policy (e.g., the expert prefers actions that lead to higher reward-to-go). We propose an algorithm named AURORAE, in short of Active preference qUeRy fOR imitAtion lEarning, which instantiates $H$ instances of AURORA, one per each time step for the finite horizon Markov Decision Process (MDP), where $H$ is the horizon. By leveraging preference-based feedback, we show that, interestingly, our algorithm can learn to outperform the

expert when the expert is suboptimal. Such a result is beyond the scope of the classic imitation learning algorithm DAGGER, and previously can only be achieved by algorithms like AGGREVATE(D) (Ross & Bagnell, 2014; Sun et al., 2017; Cheng & Boots, 2018) and LOLS (Chang et al., 2015) which require direct access to expert's actions and also reward signal – a much stronger feedback mode than ours.

To the best of our knowledge, for both contextual bandit and imitation learning with preference-based feedback, our algorithms are the first to achieve best-of-both-worlds regret bounds while at the same time minimizing query complexities.

### 1.1. Related works

**Contextual Bandits with preference feedback.** Dudík et al. (2015) is the first to consider contextual dueling bandits, and one of their algorithms achieves the optimal regret rate. Saha & Krishnamurthy (2022) studied contextual dueling bandits using a value function class and proposed an algorithm based on a reduction to online regression, which also achieves an optimal worst-case regret bound. In this paper, we mainly follow the setting of the latter and make notable improvements in two aspects: (1) in addition to the $O(\sqrt{AT})$ optimal regret rate where $A$ is the number of actions and $T$ is the number of interaction rounds, we established an instance-dependent regret upper bound that can be significantly smaller when the bandit exhibits a favorable structure; (2) our algorithm has an instance-dependent upper bound on the number of queries.

Another related work is Saha & Gaillard (2022) which achieves the best-of-both-world regret for non-contextual dueling bandits. We note that our setting is more general due to the existence of context and general function approximation, enabling us to leverage function class beyond linear and tabular cases.

**RL with preference feedback.** RL with preference feedback has been widely employed in recent advancements in AI (Ouyang et al., 2022; OpenAI, 2023). According to Wirth et al. (2017), there are generally three types of preference feedback: action preferences (Fürnkranz et al., 2012), state preferences (Wirth & Fürnkranz, 2014), and trajectory preferences (Busa-Fekete et al., 2014; Novoseller et al., 2020; Xu et al., 2020; Lee et al., 2021b; Chen et al., 2022; Saha et al., 2023; Pacchiano et al., 2021; Biyik & Sadigh, 2018; Taranovic et al.; Sadigh et al., 2017). We focus on the action preference modality with the goal of achieving tight regret bounds and query complexities.

The concurrent work from Zhan et al. (2023) investigates the experimental design in both the trajectories-based and action-based preference settings, for which they decouple

the process of collecting trajectories from querying for human feedback. Their action-based setting is the same as ours, but they mainly focus on linear parameterization, while our approach is a reduction to online regression and can leverage general function approximation beyond linear.

**Imitation learning.** In imitation learning, two common feedback modalities are typically considered: demonstrations that contain experts' actions, and preferences. The former involves directly acquiring expert actions (e.g., Ross et al. (2011); Ross & Bagnell (2014); Sun et al. (2017); Chang et al. (2015)), while the latter focuses on obtaining preferences between selected options (Chu & Ghahramani, 2005; Lee et al., 2021a; Zhu et al., 2023). Notably, Brown et al. (2019; 2020) leveraged both demonstrations and preference-based information and achieved empirical results that surpass the performance of experts. However, our imitation learning algorithm only belongs to the second category, and we established tight regret bounds and query complexities for our algorithm.

## 2. Preliminaries

In this section, we introduce the setup for contextual bandits and imitation learning with preference-based feedback. We denote $[N]$ as the set of integers between $1$ and $N$ inclusively. The set of all distributions over a set $\mathcal{S}$ is denoted by $\Delta(\mathcal{S})$.

### 2.1. Contextual Bandits with Preference-Based Feedback

In this section, we introduce the contextual dueling bandits setting. We assume a context set $\mathcal{X}$ and an action space $\mathcal{A} = [A]$. At each round $t \in [T]$, a context $x_t$ is drawn *adversarially*, and the learner's task is to decide whether they want to make a query. If they make a query, they need to select a pair of actions $(a_t, b_t) \in \mathcal{A} \times \mathcal{A}$, upon which a feedback $y_t \in \{-1, 1\}$ is revealed to the learner regarding either $a_t$ or $b_t$ is better. Specifically, we assume there is a preference function $f^\star : \mathcal{X} \times \mathcal{A} \times \mathcal{A} \to [-1, 1]$ and that the feedback $y_t$ is drawn from

$$\Pr(a_t \text{ is preferred to } b_t \,|\, x_t)$$
$$:= \Pr(y_t = 1 \,|\, x_t, a_t, b_t) = \phi\big(f^\star(x_t, a_t, b_t)\big)$$

where $\phi(d) : [-1, 1] \to [0, 1]$ is the link function, which satisfies $\phi(d) + \phi(-d) = 1$ for any $d$. If the learner does not make a query, they should still select a pair of actions $(a_t, b_t) \in \mathcal{A} \times \mathcal{A}$ but will not receive any feedback. Let $Z_t \in \{0, 1\}$ indicate whether the learner makes a query at round $t$.

We consider the realizability setting and that a general function class $\mathcal{F} \subseteq \mathcal{X} \times \mathcal{A} \times \mathcal{A} \to [-1, 1]$ is known to the learner. We suppose $f^\star$, as well as the functions in $\mathcal{F}$, is

transitive and anti-symmetric.

**Assumption 2.1.** We assume $f^\star \in \mathcal{F}$ and any functions $f \in \mathcal{F}$ satisfies the following two properties: (1) transitivity: for any $x \in \mathcal{X}$ and $a, b, c \in \mathcal{A}$, if $f(x, a, b) > 0$ and $f(x, b, c) > 0$, then we must have $f(x, a, c) > 0$; (2) anti-symmetry: $f(x, a, b) = -f(x, b, a)$ for any $x \in \mathcal{X}$ and any $a, b \in \mathcal{A}$.

We provide an example below for which Assumption 2.1 is satisfied.

*Example* 1. Assume there exists a function $r^\star : \mathcal{X} \times \mathcal{A} \to [0, 1]$ such that $f^\star(x, a, b) = r^\star(x, a) - r^\star(x, b)$ for any $x \in \mathcal{X}$ and $a, b \in \mathcal{A}$. Typically, such a function $r^\star$ represents the "reward function" of the bandit. In such a scenario, we can first parameterize a reward class $\mathcal{R} \subseteq \mathcal{X} \times \mathcal{A} \to [0, 1]$ and define $\mathcal{F} = \{f : f(x, a, b) = r(x, a) - r(x, b), r \in \mathcal{R}\}$. Moreover, it is common to have $\phi(d) := 1/(1+\exp(-d))$ in this setting, which recovers the Bradley-Terry-Luce (BTL) model (Bradley & Terry, 1952).

Assumption 2.1 ensures the existence of an optimal arm, as stated below.

**Lemma 2.2.** *Under Assumption 2.1, for any function $f \in \mathcal{F}$ and any context $x \in \mathcal{X}$, there exists an arm $a \in \mathcal{A}$ such that $f(x, a, b) \geq 0$ for any arm $b \in \mathcal{A}$. We denote this best arm by $\pi_f(x) := a$. (When the best arms are not unique, we break the tie arbitrarily and use $\pi_f(x)$ to denote any of them.)*

The learner's goal is to minimize the regret while minimizing the number of queries to the expert, which are defined as:

$$\mathrm{Regret}_T^{\mathrm{CB}} := \sum_{t=1}^{T} \big(f^\star(x_t, \pi_{f^\star}(x_t), a_t) + f^\star(x_t, \pi_{f^\star}(x_t), b_t)\big),$$

$$\mathrm{Queries}_T^{\mathrm{CB}} := \sum_{t=1}^{T} Z_t.$$

We remark that our setting generalizes that of Saha & Krishnamurthy (2022) in that we assume an additional link function $\phi$ for feedback generation, while they assume the feedback is sampled from $\Pr(y = 1 \,|\, x, a, b) = (P_t[a_t, b_t] + 1)/2$, which is captured in our setting (see Example 2). However, (Saha & Krishnamurthy, 2022) do not assume transitivity.

### 2.2. Imitation Learning with Preference-Based Feedback

For imitation learning, we consider the finite-horizon Markov decision process (MDP), which is a tuple $M(\mathcal{X}, \mathcal{A}, r, P, H)$ where $\mathcal{X}$ is the state space, $\mathcal{A}$ is the action space, $P$ is the transition kernel, $r : \mathcal{X} \times \mathcal{A} \to [0, 1]$

is the reward function, and $H$ is the length of each episode. The interaction between the learner and the environment proceeds as follows: at each episode $t \in [T]$, the learner receives an initial state $x_{t,0}$ which could be chosen adversarially. Then, the learner plays for $H$ steps. At each step $h$, the learner first decides whether they will make a query. If they make a query, they need to select a pair of actions $(a_{t,h}, b_{t,h}) \in \mathcal{A} \times \mathcal{A}$, upon which a feedback $y_{t,h} \in \{-1, 1\}$ is revealed to the learner regarding which action is preferred from the expert's perspective. Here the feedback is sampled according to

$$\Pr(a_{t,h} \text{ is preferred to } b_{t,h} \,|\, x_{t,h}, h)$$
$$:= \Pr(y_{t,h} = 1 \,|\, x_{t,h}, a_{t,h}, b_{t,h}, h) = \phi\big(f_h^\star(x, a_{t,h}, b_{t,h})\big).$$

Irrespective of whether they made a query, the learner then picks a single action from $a_{t,h}, b_{t,h}$ and transit to the next step (our algorithm will just pick an action uniformly random from $a_{t,h}, b_{t,h}$). After $H$ steps, the next episode starts. Let $Z_{t,h} \in \{0, 1\}$ indicate whether the learner decides to query at round $t$ and step $h$. We assume the function class $\mathcal{F}$ to be the product of $H$ classes, i.e., $\mathcal{F} = \mathcal{F}_0 \times \cdots \mathcal{F}_{H-1}$ where, for each $h$, we use $\mathcal{F}_h = \{f : \mathcal{X} \times \mathcal{A} \times \mathcal{A} \to [-1, 1]\}$ to model $f_h^\star$ and assume $\mathcal{F}_h$ satisfies Assumption 2.1.

A policy is a mapping $\pi : \mathcal{X} \to \Delta(\mathcal{A})$. For a policy $\pi$, the state value function for a state $x$ at step $h$ is defined as $V_h^\pi(x) := \mathbb{E}[\sum_{i=h}^{H-1} r_i \,|\, x_h = x]$ and the state-action value function for a state-action pair $(x, a)$ is $Q_h^\pi(x, a) := \mathbb{E}[\sum_{i=h}^{H-1} r_i \,|\, x_h = x, a_h = a]$. Both expectations are taken over all trajectories induced by the policy $\pi$.

We assume that different from contextual bandits, the preference function $f_h^\star$ is defined as $f_h^\star(x, a, b) := Q_h^{\pi_e}(x, a) - Q_h^{\pi_e}(x, b)$, the difference between the state-action function of an expert policy $\pi_e$. Intuitively the expert prefers actions that have high reward-to-go under its policy.

The goal is still to minimize the regret and number of queries, but the former is now defined via the performance gap:

$$\text{Regret}_T^{\text{IL}} := \sum_{t=1}^{T} \big(V_0^{\pi_e}(x_{t,0}) - V_0^{\pi_t}(x_{t,0})\big),$$

$$\text{Queries}_T^{\text{IL}} := \sum_{t=1}^{T} \sum_{h=0}^{H-1} Z_{t,h}.$$

Here $\pi_t$ is the strategy the learner used to select actions at episode $t$.

### 2.3. Link Function and Online Regression Oracle

As a standard practice (Agarwal, 2013), we assume $\phi$ is the derivative of some $\alpha$-strongly convex function $\Phi : [-1, 1] \to \mathbb{R}$ and define the associated loss function as

$\ell_\phi(d, y) = \Phi(d) - d(y + 1)/2$. In line with prior works in the literature (Foster et al., 2020; Foster & Rakhlin, 2020; Simchi-Levi & Xu, 2022; Foster et al., 2018a), our algorithm utilizes an online regression oracle and assumes it can achieve sublinear regret.

**Assumption 2.3.** We assume the learner has access to an online regression oracle pertaining to the loss $\ell_\phi$ such that for any sequence $\{(x_1, a_1, b_1, y_1), \ldots, (x_T, a_T, b_T, y_T)\}$ where the label $y_t$ is generated by $y_t \sim \phi(f^\star(x_t, a_t, b_t))$, we have $\sum_{t=1}^{T} \ell_\phi\big(f_t(x_t, a_t, b_t), y_t\big) - \inf_{f \in \mathcal{F}} \ell_\phi\big(f(x_t, a_t, b_t), y_t\big) \leq \Upsilon$ for some $\Upsilon$ that grows sublinearly with respect to $T$.

Here $\Upsilon$ represents the regret upper bound and is typically of logarithmic order in many cases (here we drop the dependence on $T$ for notation simplicity). We provide two examples to illustrate it.

*Example* 2 (Squared loss). If we consider $\Phi(d) = d^2/4 + d/2 + 1/4$, which is $1/4$-strongly convex, then we obtain $\phi(d) = (d + 1)/2$ and $\ell_\phi(d, y) = (d - y)^2/4$, thereby recovering the squared loss, which has been widely studied by prior works. For example, Rakhlin & Sridharan (2014) characterized the minimax rates for online square loss regression in terms of the offset sequential Rademacher complexity, resulting in favorable bounds for the regret. Specifically, we have $\Upsilon = O(\log |\mathcal{F}|)$ assuming the function class $\mathcal{F}$ is finite, and $\Upsilon = O(d \log(T))$ assuming $\mathcal{F}$ is a $d$-dimensional linear class. We also kindly refer the readers to Krishnamurthy et al. (2017); Foster et al. (2018a) for efficient implementations.

*Example* 3 (Logistic loss). When $\Phi(d) = \log(1 + \exp(d))$ which is strongly convex at $[-1, 1]$, we have $\phi(d) = 1/(1 + \exp(-d))$ and $\ell_\phi(d, y) = \log(1 + \exp(-yd))$. Thus, we recover the logistic regression loss, which allows us to use online logistic regression and achieve $\Upsilon = O(\log |\mathcal{F}|)$ assuming finite $\mathcal{F}$. There have been numerous endeavors in minimizing the log loss, such as Foster et al. (2018b) and Cesa-Bianchi & Lugosi (2006, Chapter 9).

## 3. Contextual Bandits with Preference-Based Active Queries

We first present our algorithm, named AURORA, for contextual dueling bandits, as shown in Algorithm 1. At each round $t \in [T]$, the online regression oracle has a predictor $f_t$, and we construct a version space $\mathcal{F}_t$ containing all functions close to $f_t$ on all observed data. Here we set the threshold $\beta := 4\Upsilon/\alpha + (16 + 24\alpha) \log \big(4\delta^{-1} \log(T)\big)/\alpha^2$, which ensures that $f^\star \in \mathcal{F}_t$ for any $t \in [T]$ with probability at least $1 - \delta$ (Lemma A.3). Thus, we have $|\mathcal{A}_t| \geq 1$ (so Line 1 will not run into trouble with high probability). We then form a candidate arm set $\mathcal{A}_t$, which consists of greedy arms induced by all functions in the version space. When $|\mathcal{A}_t| = 1$, the only arm in the set is the optimal since

**Algorithm 1** Active preference qUeRy fOR contextual bAndits (AURORA)

**Require:** function class $\mathcal{F}$, confidence parameter $\beta = \frac{4\Upsilon}{\alpha} + \frac{16+24\alpha}{\alpha^2} \log\left(4\delta^{-1}\log(T)\right)$.

1: Online regression oracle produces $f_1$.
2: **for** $t = 1, 2, \ldots, T$ **do**
3:     Receive context $x_t$.
4:     Compute the version space $\mathcal{F}_t \leftarrow$

$$\left\{ f \in \mathcal{F} : \sum_{s=1}^{t-1} Z_s \left( f(x_s, a_s, b_s) - f_t(x_s, a_s, b_s) \right)^2 \leq \beta \right\}.$$

5:     Compute the candidate arm set $\mathcal{A}_t \leftarrow \{\pi_f(x_t) : \forall f \in \mathcal{F}_t\}$.
6:     Decide whether to query $Z_t \leftarrow \mathbb{1}\{|\mathcal{A}_t| > 1\}$.
7:     **if** $Z_t = 1$ **then**
8:         $w_t \leftarrow \sup_{a,b\in\mathcal{A}_t,f,f'\in\mathcal{F}_t} f(x,a,b) - f'(x,a,b)$,
9:         $\lambda_t \leftarrow \mathbb{1}\{\sum_{s=1}^{t-1} Z_s w_s \geq \sqrt{AT/\beta}\}$.
10:        **if** $\lambda_t = 0$ **then**
11:           $p_t \leftarrow \text{Uniform}(\mathcal{A}_t)$.
12:        **else**
13:           $\gamma_t \leftarrow \sqrt{AT/\beta}$.
14:           Let $p_t$ be an arbitrary solution of $\max_{a\in\mathcal{A}_t} \sum_b f_t(x_t, a, b) p_t(b) + \frac{2}{\gamma_t p_t(a)} \leq \frac{5A}{\gamma_t}$.
15:        **end if**
16:        Sample $a_t, b_t \sim p_t$ independently and receive feedback $y_t$.
17:        Feed information $((x_t, a_t, b_t), y_t)$ to the online regression oracle which returns $f_{t+1}$.
18:     **else**
19:        Set both $a_t$ and $b_t$ as the only action in $\mathcal{A}_t$ and play them.
20:        $f_{t+1} \leftarrow f_t$.
21:     **end if**
22: **end for**

---

$f^\star \in \mathcal{F}_t$, and thus no query is needed ($Z_t = 0$). However, when $|\mathcal{A}_t| > 1$, each arm in $\mathcal{A}_t$ may be the optimal arm, and thus we need to make a query to obtain more information.

Next, we explain our strategy for making a query. Firstly, we compute $w_t$, which represents the "width" of the version space. Specifically, we proved that $w_t$ overestimated the instantaneous regret for playing any arm in $\mathcal{A}_t$ (Lemma A.2). Then, we define the indicator $\lambda_t$ that indicates if the estimated cumulative regret, $\sum_{s=1}^{t-1} Z_w w_s$, has exceeded $\sqrt{AT/\beta}$. Note that we multiply $Z_t$ to $w_t$ since we do not incur any regret when $Z_t = 0$. Below, we explain two query strategies for different values of $\lambda_t$.

- If $\lambda_t = 0$, we know that our cumulative reward has not yet exceeded $\sqrt{AT/\beta} = O(\sqrt{T})$, so we will explore as much as possible by uniform sampling from $\mathcal{A}_t$.

- If $\lambda_t = 1$, we may have incurred regret larger than $O(\sqrt{T})$, and therefore we use a technique similar to inverse gap weighting (IGW) as inspired by (Saha & Krishnamurthy, 2022) to achieve a better balance between exploration and exploitation. Specifically, we solve a simple convex program[1](Line 1), which is proved to be always feasible and the solution $p_t$ satisfies (see Lemma A.5)

$$\mathbb{E}_{a\sim p_t}\left[ f^\star(x_t, \pi_{f^\star}(x), a) \right] \leq$$
$$O\left( \gamma_t \mathbb{E}_{a,b\sim p_t}\left[ \left(f_t(x_t, a, b) - f^\star(x_t, a, b)\right)^2 \right] + \frac{A}{\gamma_t} \right). \tag{1}$$

Through this, we can convert the instantaneous regret to the point-wise error between the predictor $f_t$ and the truth $f^\star$ plus an additive $A/\gamma_t$. The cumulative point-wise error can be bounded by the online regression regret. We note that when there exists a "reward function" $r : \mathcal{X} \times \mathcal{A} \to [0,1]$ for each $f \in \mathcal{F}$ such that $f(x,a,b) = r(x,a) - r(x,b)$ (Example 1) we can define $p_t$ directly as

$$p_t(a) = \begin{cases} \frac{1}{A+\gamma_t\left(r_t(x_t,\pi_{f_t}(x_t))-r_t(x_t,a)\right)} & a \neq \pi_{f_t}(x_t) \\ 1 - \sum_{a\neq\pi_{f_t}(x_t)} p_t(a) & a = \pi_{f_t}(x_t) \end{cases},$$

where $r_t$ is the reward function associated with $f_t$, i.e., $f_t(x,a,b) = r_t(x,a) - r_t(x,b)$. This is the standard IGW expression and leads to the same guarantee as (1) (see Lemma A.6).

### 3.1. Theoretical Analysis

Towards the theoretical guarantees of Algorithm 1, we employ two quantities to characterize a contextual bandit instance: the uniform gap and the eluder dimension, which are introduced below.

**Assumption 3.1** (Uniform gap). We assume the optimal arm $\pi_{f^\star}(x)$ induced by $f^\star$ under any context $x \in \mathcal{X}$ is unique. Further, we assume a uniform gap $\Delta := \inf_x \inf_{a\neq\pi_{f^\star}(x)} f^\star(x, \pi_{f^\star}(x), a) > 0$.

We note that the existence of a uniform gap is a standard assumption in the literature of contextual bandits (Dani et al., 2008; Abbasi-Yadkori et al., 2011; Audibert et al., 2010; Garivier et al., 2019; Foster & Rakhlin, 2020; Foster et al., 2020). Next, we introduce the eluder dimension (Russo & Van Roy, 2013) and begin by defining the term "$\epsilon$-dependence".

**Definition 3.2** ($\epsilon$-dependence). Assume a function class $\mathcal{G} \subseteq \mathcal{X} \to \mathbb{R}$. We say an element $x \in \mathcal{X}$ is $\epsilon$-dependent on $\{x_1, x_2, \ldots, x_n\} \subseteq \mathcal{X}$ with respect to $\mathcal{G}$ if any pair of

---

[1]It is convex as it can be written as $|\mathcal{A}_t|$ convex constraints: $\sum_b f_t(x_t, a, b) p_t(b) + \frac{2}{\gamma_t p_t(a)} \leq \frac{5A}{\gamma_t}, \forall a \in \mathcal{A}_t$.

functions $g, g' \in \mathcal{G}$ satisfying $\sum_{i=1}^{n}(g(x_i) - g'(x_i)) \leq \epsilon^2$ also satisfies $g(x) - g'(x) \leq \epsilon$. Otherwise, we say $x$ is independent of it.

**Definition 3.3** (Eluder dimension). The $\epsilon$-eluder dimension of a function class $\mathcal{G} \subseteq \mathcal{X} \to \mathbb{R}$, denoted by $\dim_E(\mathcal{G}, \epsilon)$, is the length $d$ of the longest sequence of elements in $\mathcal{X}$ satisfying that there exists some $\epsilon' \geq \epsilon$ such that every element in the sequence is $\epsilon'$-independent of its predecessors.

Eluder dimension is one of the standard complexity measures for function classes which has been used in the literature of bandits and RL extensively (Chen et al., 2022; Osband & Van Roy, 2014; Wang et al., 2020; Foster et al., 2020; Wen & Van Roy, 2013; Jain et al., 2015; Ayoub et al., 2020; Ishfaq et al., 2021; Huang et al., 2021). Examples where eluder dimensions are small include linear functions, generalized linear models, and functions in Reproducing Kernel Hilbert Space (RKHS).

Given these quantities we are ready to state our main results. The proofs are provided in Appendix A.

**Theorem 3.4.** *Under Assumptions 2.1, 2.3 and 3.1, Algorithm 1 guarantees the following upper bounds of the regret and the number of queries:*

$$\text{Regret}_T^{\text{CB}} \leq \widetilde{O}\left(\min\left\{\sqrt{AT\beta}, \frac{A^2\beta^2\dim_E(\mathcal{F}, \Delta)}{\Delta}\right\}\right),$$

$$\text{Queries}_T^{\text{CB}} \leq \widetilde{O}\left(\min\left\{T, \frac{A^3\beta^3\dim_E^2(\mathcal{F}, \Delta)}{\Delta^2}\right\}\right)$$

*with probability at least $1-\delta$. We recall that $\beta = O(\alpha^{-1}\Upsilon + \alpha^{-2}\log(\delta^{-1}\log(T)))$. We have hidden logarithmic terms in the upper bounds for brevity.*

When the loss $\ell_\phi$ is either squared or logistic loss (Examples 2 and 3), the parameter $\beta$ will be logarithmic in $T$. In such cases, the regret is $\widetilde{O}(\min\{\sqrt{T}, \dim_E(\mathcal{F}, \Delta)/\Delta\})$ and the number of queries is $\widetilde{O}(\min\{T, \dim_E^2(\mathcal{F}, \Delta)/\Delta^2\})$, ignoring $A$ and logarithmic terms. Both consist of two components: the worst-case and the instance-dependent upper bounds. The worst-case bound provides a guarantee under all circumstances, while the instance-dependent one may significantly improve the upper bound when the problem exhibits good structures (e.g., a bounded eluder dimension and a non-trivial gap).

**Intuition of proofs.** We provide some intuition for why our algorithm has the aforementioned theoretical guarantees. First, we observe the definition of $\lambda_t$: the left term inside the indicator is non-decreasing, which allows us to divide rounds into two phases. In the first phase, $\lambda_t$ is always 0, and then at some point, it changes to 1 and remains 1 for the rest of the rounds. After realizing this, we first explain the intuition of the worst-case regret. In the first phase, as $w_t$ is an overestimate of the instantaneous regret (see

Lemma A.2), the accumulated regret in this phase cannot exceed $O(\sqrt{T})$. In the second phase, we adapt the analysis of IGW to this scenario to obtain an $O(\sqrt{T})$ upper bound. A similar technique has been used in (Saha & Krishnamurthy, 2022; Foster et al., 2020). As the regret in both phases is at most $O(\sqrt{T})$, the total regret cannot exceed $O(\sqrt{T})$. Next, we explain the intuition of instance-dependent regret. Due to the existence of a uniform gap $\Delta$, we can first prove that as long as $|\mathcal{A}_t| > 1$, we must have $w_t \geq \Delta$ (see Lemma A.1). This means that for all rounds that may incur regret, the width at that round is at least $\Delta$. However, this cannot happen too many times as this frequency is bounded by the eluder dimension, which leads to an instance-dependent regret upper bound. Leveraging a similar technique, we can also obtain an upper bound on the number of queries.

**Comparion to MINMAXDB (Saha & Krishnamurthy, 2022).** In their setting, they assume $\Pr(y = 1 \mid x, a, b) = (f^\star(x, a, b) + 1)/2$, which is a specification of our feedback model (Example 2). While our worst-case regret bound matches theirs, we improve upon their results by having an additional instance-dependent regret bound. Furthermore, we also have a guarantee on the query complexity while MINMAXDB simply queries every round.

**Comparion to ADACB (Foster et al., 2020).** Our method shares some similarities with ADACB, especially in terms of theoretical results, but differs in two aspects: (1) they assume regular contextual bandits where the learner observes the reward directly, while we assume preference feedback, and (2) they assume a stochastic setting where contexts are drawn i.i.d., but we assume the context is adversarially chosen. While these two settings may not be directly comparable, it should be noted that ADACB does not aim to minimize query complexity.

**Lower bounds.** To understand whether our algorithm attains tight upper bounds, it is important to also establish lower bounds. By using a reduction from regular multi-armed bandits to the contextual dueling bandit, we can show the following lower bounds.

**Theorem 3.5** (Lower bounds). *The following two claims hold:*

*(1) for any algorithm, there exists an instance that leads to $\text{Regret}_T^{\text{CB}} \geq \Omega(\sqrt{AT})$;*

*(2) for any algorithm achieving a worse-case expected regret upper bound in the form of $\mathbb{E}[\text{Regret}_T^{\text{CB}}] \leq O(\sqrt{AT})$, there exists an instance with gap $\Delta = \sqrt{A/T}$ that results in $\mathbb{E}[\text{Regret}_T^{\text{CB}}] \geq \Omega(A/\Delta)$ and $\mathbb{E}[\text{Queries}_T^{\text{CB}}] \geq \Omega(A/\Delta^2) = \Omega(T)$.*

By relating these lower bounds to Theorem 3.4, we can conclude that our algorithm achieves a tight dependence on the gap $\Delta$ and $T$, up to logarithmic factors, in terms of both

regret and the number of queries. Furthermore, as an additional contribution, we establish alternative lower bounds in Section A.4.1 by conditioning on the limit of regret, rather than the worst-case regret as assumed in Theorem 3.5. The lower bounds obtained in both cases are similar.

**Results without the uniform gap assumption.** We highlight that Theorem 3.4 can naturally extend to scenarios where a uniform gap does not exist (i.e., when Assumption 3.1 is not satisfied) without any modifications to the algorithm. The result is stated below, which is analogous to Theorem 3.4.

**Theorem 3.6.** *Under Assumptions 2.1 and 2.3, Algorithm 1 guarantees the following upper bounds of the regret and the number of queries:*

$$\text{Regret}_T^{\text{CB}}$$
$$\leq \widetilde{O}\left(\min_{\epsilon > 0} \min\left\{\sqrt{AT\beta}, T_\epsilon \beta + \frac{A^2 \beta^2 \text{dim}_E(\mathcal{F}, \epsilon)}{\epsilon}\right\}\right),$$
$$\text{Queries}_T^{\text{CB}}$$
$$\leq \widetilde{O}\left(\min_{\epsilon > 0} \min\left\{T, T_\epsilon^2 \beta / A + \frac{A^3 \beta^3 \text{dim}_E^2(\mathcal{F}, \epsilon)}{\epsilon^2}\right\}\right)$$

*with probability at least $1 - \delta$. Here we define $T_\epsilon := \sum_{t=1}^{T} \mathbb{1}\{\text{Gap}(x_t) \leq \epsilon\}$ where we denote the gap under a specific context $x$ as $\text{Gap}(x) := \min_{a \neq \pi_{f^\star}(x)} f^\star(x, \pi_{f^\star}(x), a)$. We have hidden logarithmic terms in the upper bounds for brevity.*

Compared to Theorem 3.4, it differs in the extra term involving $T_\epsilon$. Here $\epsilon$ denotes a gap threshold, and $T_\epsilon$ measures how many times the context falls into a small-gap region. We highlight that $T_\epsilon$ is small under certain conditions such as the Tsybakov noise condition (Tsybakov, 2004). It is also worth mentioning that our algorithm is agnostic to $\epsilon$, thus allowing us to take the minimum over all $\epsilon > 0$.

# 4. Imitation Learning with Preference-Based Active Queries

In this section, we introduce our second algorithm, which is presented in Algorithm 2. In essence, we treat the MDP as a concatenation of $H$ contextual bandits and run an AURORA (Algorithm 1) for each time step. Specifically, we first create $H$ instances of AURORA, denoted by AURORA$_h$ ($h = 0, \ldots, H - 1$). Here we consider AURORA as an interactive program that takes the context $x_t$ as input and can output $a_t$, $b_t$, and $Z_t$. At each episode $t$ and each time step $h$ therein, we first feed the current state $x_{t,h}$ to AURORA$_h$ as the context; then, AURORA$_h$ will decide whether to query. If it decides to make a query, we will ask for the feedback $y_{t,h}$ on the proposed two actions $a_{t,h}, b_{t,h}$, and let AURORA$_h$ perform an update with the information $((x_{t,h}, a_{t,h}, b_{t,h}), y_{t,h})$. We re-

---

**Algorithm 2** Active preference qUeRy fOR imitAtion lEarning (AURORAE)

**Require:** function class $\mathcal{F}_0, \mathcal{F}_1, \ldots, \mathcal{F}_{H-1}$, confidence parameter $\beta$.
1: Create $H$ instances of Algorithm 1: AURORA$_h(\mathcal{F}_h, \beta)$ for $h = 0, 1, \ldots, H - 1$.
2: **for** $t = 1, 2, \ldots, T$ **do**
3:     Receive initial state $x_{t,0}$.
4:     **for** $h = 0, 1, \ldots, H - 1$ **do**
5:         Feed $x_{t,h}$ to AURORA$_h(\mathcal{F}_h, \beta)$.
6:         Get $a_{t,h}, b_{t,h}, Z_{t,h}$ from AURORA$_h(\mathcal{F}_h, \beta)$.
7:         **if** $Z_{t,h} = 1$ **then**
8:             Receive feedback $y_{t,h}$.
9:             Feed information $((x_{t,h}, a_{t,h}, b_{t,h}), y_{t,h})$ to AURORA$_h(\mathcal{F}_h, \beta)$ to perform an update.
10:        **end if**
11:        Execute $a \sim \text{Uniform}(\{a_{t,h}, b_{t,h}\})$ and transits to $x_{t,h+1}$.
12:     **end for**
13: **end for**

---

call that the noisy binary feedback $y_{t,h}$ is sampled as $y_{t,h} \sim \phi(Q_h^{\pi_e}(x_{t,h}, a_{t,h}) - Q_h^{\pi_e}(x_{t,h}, b_{t,h}))$ in this case. We emphasize that the learner has no access to $a \sim \pi_e(x_{t,h})$ like DAGGER (Ross et al., 2011) nor reward-to-go access like AGGREVATE(D) (Ross & Bagnell, 2014; Sun et al., 2017). Finally, the learner executes one of the actions, and the state transits to $x_{t,h+1}$. We repeat the same process for AURORA$_{h+1}$, until the end of the episode. We name this algorithm AURORAE, the plural form of AURORA, which signifies that the algorithm is essentially a stack of multiple AURORA instances.

## 4.1. Theoretical analysis

As Algorithm 2 is essentially a stack of Algorithm 1, we can inherit many of the theoretical guarantees from the previous section. To state the results, we first extend Assumption 3.1 into imitation learning.

**Assumption 4.1** (Uniform Gap). For all $h$, we assume the optimal action for $f_h^\star$ under any state $x \in \mathcal{X}$ is unique. Further, we assume a uniform gap $\Delta := \inf_h \inf_x \inf_{a \neq \pi_{f_h^\star}(x)} f_h^\star(x, \pi_{f_h^\star}(x), a) > 0$.

We remark that, just as Assumption 3.1 is a common condition in the bandit literature, Assumption 4.1 is also common in MDPs (Du et al., 2019; Foster et al., 2020; Simchowitz & Jamieson, 2019; Jin & Luo, 2020; Lykouris et al., 2021; He et al., 2021). The theoretical guarantee for Algorithm 2 is presented in Theorem 4.2. We note a technical difference between this result and Theorem 3.4: although we treat the MDP as a concatenation of $H$ contextual bandits, the instantaneous regret of imitation learning is defined as the

performance gap between the combined policy derived from the $H$ instances as a cohesive unit and the optimal policy. This necessitates the use of performance difference lemma (Lemma B.4) to get a unified result.

**Theorem 4.2.** *Under Assumptions 2.1, 2.3 and 4.1, Algorithm 2 guarantees the following upper bounds of the regret and the number of queries:*

$$\text{Regret}_T^{\text{IL}}$$

$$\leq \widetilde{O}\left(H \cdot \min\left\{\sqrt{AT\beta}, \frac{A^2\beta^2\dim_E(\mathcal{F},\Delta)}{\Delta}\right\}\right) - \text{Adv}_T,$$

$$\text{Queries}_T^{\text{IL}} \leq \widetilde{O}\left(H \cdot \min\left\{T, \frac{A^3\beta^3\dim_E^2(\mathcal{F},\Delta)}{\Delta^2}\right\}\right)$$

*with probability at least $1 - \delta$. Here $\text{Adv}_T \coloneqq \sum_{t=1}^{T}\sum_{h=0}^{H-1}\mathbb{E}_{x_{t,h}\sim d_{x_{t,0},h}^{\pi_t}}[\max_a A_h^{\pi_e}(x_{t,h},a)] \geq 0$, and $d_{x_{t,0},h}^{\pi_t}(x)$ denotes the probability of $\pi_t$ [2] reaching the state $x$ at time step $h$ starting from inital state $x_{t,0}$. We also recall that $\beta = O(\alpha^{-1}\Upsilon + \alpha^{-2}\log(H\delta^{-1}\log(T)))$. Here we have hidden logarithmic terms in the upper bounds for brevity.*

Compared to Theorem 3.4, the main terms of the upper bounds for imitation learning are precisely the bounds in Theorem 3.4 multiplied by $H$. In the proof presented in Appendix A.6, we use the performance difference lemma to reduce the regret of imitation learning to the sum of the regret of $H$ contextual dueling bandits, which explains this factor of $H$.

Another interesting point is that the main term of the regret upper bound is subtracted by a non-negative term $\text{Adv}_T$, which measures the degree to which we can *outperform* the expert policy. This means that our algorithm not only competes with the expert policy but can also surpass it to some extent. This guarantee is stronger than that of DAGGER (Ross et al., 2011) in that DAGGER cannot ensure the learned policy is better than the expert policy regardless of how suboptimal the expert may be. While this may look surprising at first glance since we are operating under a somewhat weaker query mode than that of DAGGER, we note that by querying experts for comparisons on pairs of actions with feedback sampling as $y \sim \phi(Q^{\pi_e}(x,a) - Q^{\pi_e}(x,b))$, it is possible to identify the action that maximizes $Q^{\pi_e}(x,a)$. Finally, we remark that our worst-case regret bound is similar to that of Ross & Bagnell (2014); Sun et al. (2017), which can also outperform a suboptimal expert but require access to both expert's actions and reward signals – a much stronger query model than ours.

---

[2]Policy $\pi_t$ consists of $H$ time-dependent policies $\pi_{t,1},\ldots,\pi_{t,H}$, where each $\pi_{t,h}$ is defined implicitly via AURORA$_h$, i.e., $\pi_{t,h}$ generates action as follows: given $x_{t,h}$, AURORA$_h$ recommends $a_{t,h}, b_{t,h}$, followed by uniformly sampling an action from $\{a_{t,h}, b_{t,h}\}$.

## 5. Discussion and Future Work

We presented interactive decision-making algorithms that learn from preference-based feedback while minimizing query complexity. Our algorithms for contextual bandits and imitation learning share worst-case regret bounds similar to the bounds of the state-of-art algorithms in standard settings while maintaining instance-dependent regret bounds and query complexity bounds. Notably, our imitation learning algorithm can outperform suboptimal experts, matching the result of (Ross & Bagnell, 2014; Sun et al., 2017), which operates under much stronger feedback.

In terms of future work, we believe our result on contextual dueling bandits can be extended to the stochastic setting where we may replace the eluder dimension with the value function disagreement coefficient (Foster et al., 2020), which is weaker than the eluder dimension, and replace the online regression oracle by a supervised-learning batch regression oracle. We also conjecture that the dependence on the eluder dimension in the query complexity bound can be improved. Finally, another interesting direction is to develop practical implementations of our proposed algorithms.

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

# A. Missing Proofs

## A.1. Supporting Lemmas

**Lemma A.1.** *For any $t \in [T]$, if $f^\star \in \mathcal{F}_t$, then we have $w_t \geq \Delta$ whenever $|\mathcal{A}_t| > 1$.*

*Proof of Lemma A.1.* When $|\mathcal{A}_t| > 1$, we know there exists a function $f' \in \mathcal{F}_t$ satisfying

$$a' := \pi_{f'}(x_t) \neq \pi_{f^\star}(x_t) =: a_t^\star.$$

Then we have $\Delta \leq f^\star(x_t, a_t^\star, a') \leq f^\star(x_t, a_t^\star, a') - f'(x_t, a_t^\star, a') \leq w_t$ where the second inequality holds since $f'(x_t, a_t^\star, a') \leq 0$. $\qquad\square$

**Lemma A.2.** *For any $t \in [T]$ and any arm $a \in \mathcal{A}_t$, we have $f^\star(x_t, \pi_{f^\star}(x_t), a) \leq w_t$.*

*Proof of Lemma A.2.* For any $a \in \mathcal{A}_t$, by the definition of $\mathcal{A}_t$, there must exists a function $f$ for which $a = \pi_f(x_t)$. Hence,

$$f^\star(x_t, \pi_{f^\star}(x_t), a) \leq f^\star(x_t, \pi_{f^\star}(x_t), a) - f(x_t, \pi_{f^\star}(x_t), a) \leq w_t,$$

where the first inequality holds since $f(x_t, \pi_{f^\star}(x_t), a) \leq 0$. $\qquad\square$

**Lemma A.3.** *The following holds with probability at least $1 - \delta$ for any $T > 3$,*

$$\sum_{t=1}^{T} Z_t \big(f^\star(x_t, a_t, b_t) - f_t(x_t, a_t, b_t)\big)^2 \leq \frac{4\Upsilon}{\alpha} + \frac{16 + 24\alpha}{\alpha^2} \log\big(4\delta^{-1} \log(T)\big).$$

*Proof of Lemma A.3.* Throughout the proof, we denote $z_t := (x_t, a_t, b_t)$ for notational simplicity. We define $D_\Phi$ as the Bregman divergence of the function $\Phi$:

$$D_\Phi(u, v) = \Phi(u) - \Phi(v) - \phi(v)(u - v)$$

where we recall that $\phi = \Phi'$ is the derivative of $\Phi$. Since $\Phi$ is $\alpha$-strong convex, we have $\alpha(u - v)^2/2 \leq D_\Phi(u, v)$, and hence,

$$\sum_{t=1}^{T} Z_t \big(f^\star(z_t) - f_t(z_t)\big)^2 \leq \frac{2}{\alpha} \sum_{t=1}^{T} Z_t D_\Phi(f_t(z_t), f^\star(z_t)). \tag{2}$$

Hence, it suffice derive an upper bound for the Bregman divergence then. Define $\nu_t$ as below:

$$
\begin{aligned}
\nu_t :=& Z_t \Big[ D_\Phi \left( f_t(z_t), f^\star(z_t) \right) - \left( \ell \left( f_t(z_t), y_t \right) - \ell \left( f^\star(z_t), y_t \right) \right) \Big] \\
=& Z_t \Big[ D_\Phi \left( f_t(z_t), f^\star(z_t) \right) - \left( \Phi \left( f_t(z_t) \right) - (y_t + 1)f_t(z_t)/2 - \Phi \left( f^\star(z_t) \right) + (y_t + 1)f^\star(z_t)/2 \right) \Big] \\
=& Z_t \Big[ \Phi \left( f_t(z_t) \right) - \Phi \left( f^\star(z_t) \right) - \phi \left( f^\star(z_t) \right) \left( f_t(z_t) - f^\star(z_t) \right) \\
& \quad - \left( \Phi \left( f_t(z_t) \right) - (y_t + 1)f_t(z_t)/2 - \Phi \left( f^\star(z_t) \right) + (y_t + 1)f^\star(z_t)/2 \right) \Big] \\
=& Z_t \big( f_t(z_t) - f^\star(z_t) \big) \big( (y_t + 1)/2 - \phi(f^\star(z_t)) \big)
\end{aligned}
$$

We note that $\mathbb{E}_t[(y_t + 1)/2] = \phi(f^\star(z_t))$, and thus $\mathbb{E}_t[\nu_t] = 0$, which means $\nu_t$ is a martingale difference sequence. Now we bound the value and the conditional variance of $\nu_t$ in order to derive concentration results.

1. Bound the value of $\nu_t$:

$$|\nu_t| \leq |(y_t + 1)/2 - \phi\left( f^\star(z_t) \right)| \cdot |f_t(z_t) - f^\star(z_t)|_\infty \leq 1 \cdot 2 = 2.$$

2. Bound the conditional variance of $\nu_t$:

$$
\begin{aligned}
\mathbb{E}_t[\nu_t^2] &= Z_t \, \mathbb{E}_t\left[ \left( (y_t + 1)/2 - \phi\left(f^\star(z_t)\right) \right)^2 \left( f_t(z_t) - f^\star(z_t) \right)^2 \right] \\
&\leq Z_t \, \mathbb{E}_t\left[ \left( f_t(z_t) - f^\star(z_t) \right)^2 \right] \\
&\leq Z_t \, \mathbb{E}_t\left[ \frac{2}{\alpha} \cdot D_\Phi(f_t(z_t), f^\star(z_t)) \right] \\
&\leq \frac{2Z_t}{\alpha} D_\Phi(f_t(z_t), f^\star(z_t))
\end{aligned}
$$

where for the last line we note that $x_t, g_t$ are measurable at $t$.

Now we apply Lemma B.1, which yields for any $\delta < 1/e$ and $T > 3$, with probability at least $1 - 4\delta \log(T)$,

$$
\begin{aligned}
\sum_{t=1}^T \nu_t &\leq \max\left\{ 2\sqrt{\sum_{t=1}^T \frac{2Z_t}{\alpha} D_\Phi(f_t(z_t), f^\star(z_t))}, 6\sqrt{\log(1/\delta)} \right\} \sqrt{\log(1/\delta)} \\
&\leq 2\sqrt{\sum_{t=1}^T \frac{2Z_t}{\alpha} D_\Phi(f_t(z_t), f^\star(z_t)) \log(1/\delta)} + 6\log(1/\delta) \qquad \text{(since } \max(a,b) \leq a + b) \\
&\leq \sum_{t=1}^T \frac{1}{2} Z_t D_\Phi(f_t(z_t), f^\star(z_t)) + \frac{4\log(1/\delta)}{\alpha} + 6\log(1/\delta) \qquad \text{(AM-GM)}
\end{aligned}
$$

Recall the definition of $\nu_t$, and we conclude that

$$
\sum_{t=1}^T Z_t D_\Phi(f_t(z_t), f^\star(z_t)) - \sum_{t=1}^T Z_t\left( \ell_\phi\left(f_t(z_t), y_t\right) - \ell_\phi\left(f^\star(z_t), y_t\right) \right) \leq
$$
$$
\sum_{t=1}^T \frac{1}{2} Z_t D_\Phi(f_t(z_t), f^\star(z_t)) + \frac{4\log(1/\delta)}{\alpha} + 6\log(1/\delta),
$$

which implies

$$
\frac{1}{2}\sum_{t=1}^T Z_t D_\Phi(f_t(z_t), f^\star(z_t)) \leq \sum_{t=1}^T Z_t\left( \ell_\phi\left(f_t(z_t), y_t\right) - \ell_\phi\left(f^\star(z_t), y_t\right) \right) + \frac{4\log(1/\delta)}{\alpha} + 6\log(1/\delta).
$$

Plugging this upper bound of Bregman divergence into (2), we obtain that, with probability at least $1 - 4\delta \log(T)$, for any $\delta < 1/e$ and $T > 3$, we have

$$
\sum_{t=1}^T Z_t\left( f^\star(z_t) - f_t(z_t) \right)^2 \leq \frac{4}{\alpha}\Upsilon + \left( \frac{16}{\alpha^2} + \frac{24}{\alpha} \right) \log(\delta^{-1}) =: \beta
$$

Finally, we finish the proof by adjusting the coefficient $\delta$ to obtain the desired result. $\qquad \square$

The following lemma is a variant of Russo & Van Roy (2013, Proposition 3), with the main difference being that (1) the version space is established using the function produced by the oracle instead of the least squares estimator, and (2) the extra multiplicative factor $Z_t$.

**Lemma A.4.** *For Algorithm 1, it holds that*

$$
\sum_{t=1}^T Z_t \mathbb{1}\left\{ \sup_{f, f' \in \mathcal{F}_t} f(x_t, a_t, b_t) - f'(x_t, a_t, b_t) > \epsilon \right\} \leq \left( \frac{4\beta}{\epsilon^2} + 1 \right) \dim_E(\mathcal{F}, \epsilon) \tag{3}
$$

*for any constant $\epsilon > 0$,*

*Proof of Lemma A.4.* We first define a subsequence consisting only of the elements for which we made a query in that round. Specifically, we define $((x_{i_1}, a_{i_1}, b_{i_1}), (x_{i_2}, a_{i_2}, b_{i_2}), \ldots, (x_{i_k}, a_{i_k}, b_{i_k}))$ where $1 \leq i_1 < i_2 < \cdots < i_k \leq T$ and $(x_t, a_t, b_t)$ belongs to the subsequence if and only if $Z_t = 1$. We further simplify the notation by defining $z_j := (x_{i_j}, a_{i_j}, b_{i_j})$ and $f(z_j) := f(x_{i_j}, a_{i_j}, b_{i_j})$. Then we note that the left-hand side of (3) is equivalent to

$$\sum_{j=1}^{k} \mathbb{1} \left\{ \sup_{f, f' \in \mathcal{F}_j} f(z_j) - f'(z_j) > \epsilon \right\}, \tag{4}$$

and the version space in Algorithm 1 is equal to

$$\mathcal{F}_j = \left\{ f \in \mathcal{F} : \sum_{s=1}^{j-1} \left( f(z_s) - f_t(z_s) \right)^2 \leq \beta \right\}. \tag{5}$$

Hence, it suffice to establish the lower bound for (4) under the version space of (5). To that end, we make one more simplicication in notation: we denote

$$w'_j := \sup_{f, f' \in \mathcal{F}_j} f(z_j) - f'(z_j)$$

We begin by showing that if $w'_j > \epsilon$ for some $j \in [k]$, then $z_j$ is $\epsilon$-dependent on at most $4\beta/\epsilon^2$ disjoint subsequence of its predecesors. To see this, we note that when $w'_j > \epsilon$, there must exist two function $f, f' \in \mathcal{F}_j$ such that $f(z_j) - f'(z_j) > \epsilon$. If $z_j$ is $\epsilon$-dependent on a subsequence $(z_{i_1}, z_{i_2}, \ldots, z_{i_n})$ of its predecessors, we must have

$$\sum_{s=1}^{n} \left( f(z_{i_s}) - f'(z_{i_s}) \right)^2 > \epsilon^2.$$

Hence, if $z_j$ is $\epsilon$-dependent on $l$ disjoint subsequences, we have

$$\sum_{s=1}^{j-1} \left( f(z_s) - f'(z_s) \right)^2 > l\epsilon^2. \tag{6}$$

For the left-hand side above, we also have

$$\sum_{s=1}^{j-1} \left( f(z_s) - f'(z_s) \right)^2 \leq 2 \sum_{s=1}^{j-1} \left( f(z_s) - f_t(z_s) \right)^2 + 2 \sum_{s=1}^{j-1} \left( f_t(z_s) - f'(z_s) \right)^2 \leq 4\beta \tag{7}$$

where the first inequality holds since $(a + b)^2 \leq 2(a^2 + b^2)$ for any $a, b$, and the second inequality holds by (5). Combining (6) and (7), we get that $l \leq 4\beta/\epsilon^2$.

Next, we show that for any sequence $(z'_1, \ldots, z'_\tau)$, there is at least one element that is $\epsilon$-dependent on at least $\tau/d - 1$ disjoint subsequence of its predecessors, where $d := \dim_E(\mathcal{F}, \epsilon)$. To show this, let $m$ be the integer satisfying $md + 1 \leq \tau \leq md + d$. We will construct $m$ disjoint subsequences, $B_1, \ldots, B_m$. At the beginning, let $B_i = (z'_i)$ for $i \in [m]$. If $z'_{m+1}$ is $\epsilon$-dependent on each subsequence $B_1, \ldots, B_m$, then we are done. Otherwise, we select a subsequence $B_i$ which $z'_{m+1}$ is $\epsilon$-independent of and append $z'_{m+1}$ to $B_i$. We repeat this process for all elements with indices $j > m + 1$ until either $z'_j$ is $\epsilon$-dependent on each subsequence or $j = \tau$. For the latter, we have $\sum_{i=1}^{m} |B_i| \geq md$, and since each element of a subsequence $B_i$ is $\epsilon$-independent of its predecesors, we must have $|B_i| = d$ for all $i$. Then, $z_\tau$ must be $\epsilon$-dependent on each subsequence by the definition of eluder dimension.

Finally, let's take the sequence $(z'_1, \ldots z'_\tau)$ to be the subsequence of $(z_1, \ldots, z_k)$ consisting of elements $z_j$ for which $w'_j > \epsilon$. As we have established, we have (1) each $z'_j$ is $\epsilon$-dependent on at most $4\beta/\epsilon^2$ disjoint subsequences, and (2) some $z'_j$ is $\epsilon$-dependent on at least $\tau/d - 1$ disjoint subsequences. Therefore, we must have $\tau/d - 1 \leq 4\beta/\epsilon^2$, implying that $\tau \leq (4\beta/\epsilon^2 + 1)d$. $\square$

The following lemma is adopted from Saha & Krishnamurthy (2022, Lemma 3).

**Lemma A.5.** *For any function $f \in \mathcal{F}$ and any context $x \in \mathcal{X}$, the following convex program of $p \in \Delta(\mathcal{A})$ is always feasible:*

$$\forall a \in \mathcal{A} : \sum_b f(x,a,b)p(b) + \frac{2}{\gamma p(a)} \leq \frac{5A}{\gamma}.$$

*Furthermore, any solution $p$ satisfies:*

$$\mathop{\mathbb{E}}_{a \sim p} \left[ f^\star(x, \pi_{f^\star}(x), a) \right] \leq \frac{\gamma}{4} \mathop{\mathbb{E}}_{a,b \sim p} \left[ \left( f(x,a,b) - f^\star(x,a,b) \right)^2 \right] + \frac{5A}{\gamma}$$

*whenever $\gamma \geq 2A$.*

**Lemma A.6.** *Assume that for each $f \in \mathcal{F}$, there exists an associated function $r : \mathcal{X} \times \mathcal{A} \to [0,1]$ such that $f(x,a,b) = r(x,a) - r(x,b)$ for any $x \in \mathcal{X}$ and $a,b \in \mathcal{A}$. In this case, for any context $x \in \mathcal{X}$, if we define $p$ as*

$$p(a) = \begin{cases} \frac{1}{A + \gamma\left(r(x,\pi_f(x)) - r(x,a)\right)} & a \neq \pi_f(x) \\ 1 - \sum_{a \neq \pi_f(x)} p(a) & a = \pi_f(x) \end{cases},$$

*then we have*

$$\mathop{\mathbb{E}}_{a \sim p} \left[ f^\star(x, \pi_{f^\star}(x), a) \right] \leq \gamma \mathop{\mathbb{E}}_{a,b \sim p} \left[ \left( f(x,a,b) - f^\star(x,a,b) \right)^2 \right] + \frac{A}{\gamma}$$

*Proof of Lemma A.6.* Fix any $b \in \mathcal{A}$. Then, the distribution $p$ can be rewritten as

$$p(a) = \begin{cases} \left( A + 2\gamma \left( \frac{r(x,\pi_f(x)) - r(x,b) + 1}{2} - \frac{r(x,a) - r(x,b) + 1}{2} \right) \right)^{-1} & a \neq \pi_f(x) \\ 1 - \sum_{a \neq \pi_f(x)} p(a) & a = \pi_f(x) \end{cases}.$$

Therefore, denoting $f^\star(x,a,b) = r^\star(x,a) - r^\star(x,b)$ for some function $r^\star$, we have

$$\begin{aligned}
\mathop{\mathbb{E}}_{a \sim p} \left[ f^\star(x, \pi_{f^\star}(x), a) \right] &= \mathop{\mathbb{E}}_{a \sim p} \left[ r^\star(x, \pi_{f^\star}(x)) - r^\star(x,a) \right] \\
&= 2 \mathop{\mathbb{E}}_{a \sim p} \left[ \frac{r^\star(x, \pi_{f^\star}(x)) - r^\star(x,b) + 1}{2} - \frac{r^\star(x,a) - r^\star(x,b) + 1}{2} \right] \\
&\leq 2 \cdot 2\gamma \mathop{\mathbb{E}}_{a \sim p} \left[ \left( \frac{r(x,a) - r(x,b) + 1}{2} - \frac{r^\star(x,a) - r^\star(x,b) + 1}{2} \right)^2 \right] + \frac{A}{\gamma} \\
&= \gamma \mathop{\mathbb{E}}_{a \sim p} \left[ \left( f(x,a,b) - f^\star(x,a,b) \right)^2 \right] + \frac{A}{\gamma}
\end{aligned}$$

where for the inequality above we invoked Lemma B.2 with $\hat{y}(a) = (r(x,a) - r(x,b) + 1)/2$ and $y^\star(a) = (r^\star(x,a) - r^\star(x,b) + 1)/2$. We note that the above holds for any $b \in \mathcal{A}$. Hence, we complete the proof by sampling $b \sim p$. $\square$

**Lemma A.7.** *Assume $f^\star \in \mathcal{F}_t$ for all $t \in [T]$. Suppose there exists some $t' \in [T]$ such that $\lambda_t = 0$ for all $t \leq t'$. Then we have*

$$\sum_{t=1}^{t'} Z_t w_t \leq 56 A^2 \beta \cdot \frac{\dim_E (\mathcal{F}, \Delta)}{\Delta} \cdot \log(2/(\delta\Delta))$$

*with probability at least $1 - \delta$.*

*Proof.* Since $f^\star \in \mathcal{F}_t$, we always have $\pi_{f^\star}(x_t) \in \mathcal{A}_t$ for all $t \in [T]$. Hence, whenever $Z_t$ is zero, we have $\mathcal{A}_t = \{\pi_{f^\star}(x_t)\}$ and thus we do not incur any regret. Hence, we know $Z_t w_t$ is either $0$ or at least $\Delta$ by Lemma A.1. Let us fix an integer

$m > 1/\Delta$, whose value will be specified later. We divide the interval $[\Delta, 1]$ into bins of width $1/m$ and conduct a refined study of the sum of $Z_t w_t$:

$$
\begin{aligned}
\sum_{t=1}^{t'} Z_t w_t &\leq \sum_{t=1}^{t'} \sum_{j=0}^{(1-\Delta)m-1} Z_t w_t \cdot \mathbb{1}\left\{ Z_t w_t \in \left[\Delta + \frac{j}{m}, \Delta + \frac{j+1}{m}\right]\right\} \\
&\leq \sum_{j=0}^{(1-\Delta)m-1} \left(\Delta + \frac{j+1}{m}\right) \sum_{t=1}^{t'} Z_t \mathbb{1}\left\{ w_t \geq \Delta + \frac{j}{m}\right\} \\
&= \sum_{j=0}^{(1-\Delta)m-1} \left(\Delta + \frac{j+1}{m}\right) \sum_{t=1}^{t'} Z_t \mathbb{1}\left\{ \sup_{a,b\in\mathcal{A}_t} \sup_{f,f'\in\mathcal{F}_t} f(x_t,a,b) - f'(x_t,a,b) \geq \Delta + \frac{j}{m}\right\} \\
&= \sum_{j=0}^{(1-\Delta)m-1} \left(\Delta + \frac{j+1}{m}\right) \sum_{t=1}^{t'} Z_t \sup_{a,b\in\mathcal{A}_t} \mathbb{1}\left\{ \sup_{f,f'\in\mathcal{F}_t} f(x_t,a,b) - f'(x_t,a,b) \geq \Delta + \frac{j}{m}\right\} \\
&\leq \sum_{j=0}^{(1-\Delta)m-1} \left(\Delta + \frac{j+1}{m}\right) \sum_{t=1}^{t'} Z_t \sum_{a,b} \mathbb{1}\left\{ \sup_{f,f'\in\mathcal{F}_t} f(x_t,a,b) - f'(x_t,a,b) \geq \left(\Delta + \frac{j}{m}\right)\right\} \\
&\leq \sum_{j=0}^{(1-\Delta)m-1} \left(\Delta + \frac{j+1}{m}\right) A^2 \underbrace{\sum_{t=1}^{t'} Z_t \mathop{\mathbb{E}}_{a,b\sim p_t} \mathbb{1}\left\{ \sup_{f,f'\in\mathcal{F}_t} f(x_t,a,b) - f'(x_t,a,b) \geq \left(\Delta + \frac{j}{m}\right)\right\}}_{(*)}
\end{aligned}
$$

where in the third inequality we replace the supremum over $a,b$ by the summation over $a,b$, and in the last inequality we further replace it by the expectation. Here recall that $p_t(a)$ is uniform when $\lambda_t = 0$, leading to the extra $A^2$ factor. To deal with $(*)$, we first apply Lemma B.3 to recover the empirical $a_t$ and $b_t$, and then apply Lemma A.4 to get an upper bound via the eluder dimension:

$$
\begin{aligned}
(*) &\leq 2 \sum_{t=1}^{t'} Z_t \mathbb{1}\left\{ \sup_{f,f'\in\mathcal{F}_t} f(x_t,a_t,b_t) - f'(x_t,a_t,b_t) \geq \left(\Delta + \frac{j}{m}\right)\right\} + 8\log(\delta^{-1}) \\
&\leq 2\left(\frac{4\beta}{\left(\Delta + \frac{j}{m}\right)^2} + 1\right) \dim_E(\mathcal{F};\Delta) + 8\log(\delta^{-1}) \\
&\leq \frac{10\beta}{\left(\Delta + \frac{j}{m}\right)^2} \cdot \dim_E(\mathcal{F};\Delta) + 8\log(\delta^{-1})
\end{aligned}
$$

with probability at least $1 - \delta$. Plugging $(*)$ back, we obtain

$$
\sum_{t=1}^{t'} Z_t w_t \leq \sum_{j=0}^{(1-\Delta)m-1} \left( \Delta + \frac{j+1}{m} \right) \cdot \frac{10A^2\beta}{\left( \Delta + \frac{j}{m} \right)^2} \cdot \dim_E(\mathcal{F}; \Delta) + 8mA^2 \log(\delta^{-1})
$$

$$
= 10A^2\beta \cdot \dim_E(\mathcal{F}, \Delta) \sum_{j=0}^{(1-\Delta)m-1} \frac{\Delta + \frac{j+1}{m}}{\left( \Delta + \frac{j}{m} \right)^2} + 8mA^2 \log(\delta^{-1})
$$

$$
\leq 10A^2\beta \cdot \dim_E(\mathcal{F}, \Delta) \left( \frac{\Delta + 1/m}{\Delta^2} + \sum_{j=1}^{(1-\Delta)m-1} \frac{2}{\Delta + \frac{j}{m}} \right) + 8mA^2 \log(\delta^{-1})
$$

$$
\leq 10A^2\beta \cdot \dim_E(\mathcal{F}, \Delta) \sum_{j=0}^{(1-\Delta)m-1} \frac{2}{\Delta + \frac{j}{m}} + 8mA^2 \log(\delta^{-1})
$$

$$
\leq 20A^2\beta \cdot \dim_E(\mathcal{F}, \Delta) \sum_{j=0}^{(1-\Delta)m-1} \int_{j-1}^{j} \frac{1}{\Delta + \frac{x}{m}} \, dx + 8mA^2 \log(\delta^{-1})
$$

$$
= 20A^2\beta \cdot \dim_E(\mathcal{F}, \Delta) \int_{-1}^{(1-\Delta)m-1} \frac{1}{\Delta + \frac{x}{m}} \, dx + 8mA^2 \log(\delta^{-1})
$$

$$
= 20A^2\beta \cdot \dim_E(\mathcal{F}, \Delta) \cdot m \log\left( \frac{1}{\Delta - m^{-1}} \right) + 8mA^2 \log(\delta^{-1})
$$

where for the second inequality, we use the fact that $(j+1)/m \leq 2j/m$ for any $j \geq 1$; for the third inequality, we assume $m > 1/\Delta$. Setting $m = 2/\Delta$, we arrive at

$$
\sum_{t=1}^{t'} Z_t w_t \leq 40A^2\beta \cdot \frac{\dim_E(\mathcal{F}, \Delta)}{\Delta} \cdot \log(2/\Delta) + 16A^2 \log(\delta^{-1})/\Delta
$$

$$
\leq 56A^2\beta \cdot \frac{\dim_E(\mathcal{F}, \Delta)}{\Delta} \cdot \log(2/(\delta\Delta)),
$$

which completes the proof. $\qquad\square$

**Lemma A.8.** *Whenever*

$$
56A^2\beta \cdot \dim_E(\mathcal{F}, \Delta) \cdot \log(2/(\delta\Delta))/\Delta < \sqrt{AT/\beta},
$$

*we have $\lambda_1 = \lambda_2 = \cdots = \lambda_T = 0$ with probability at least $1 - \delta$.*

*Proof of Lemma A.8.* We prove it via contradiction. Assume the inequality holds but there exists $t'$ for which $\lambda_{t'} = 1$. Without loss of generality, we assume that $\lambda_t = 0$ for all $t < t'$, namely that $t'$ is the first time that $\lambda_t$ is 1. Then by definition of $\lambda_{t'}$, we have

$$
\sum_{s=1}^{t'-1} Z_s w_s \geq \sqrt{AT/\beta}.
$$

On the other hand, by Lemma A.7, we have

$$
\sum_{s=1}^{t'-1} Z_s w_s \leq 56A^2\beta \cdot \frac{\dim_E(\mathcal{F}, \Delta)}{\Delta} \cdot \log(2/(\delta\Delta)).
$$

The combination of the above two inequalities contradicts with the conditions. $\qquad\square$

### A.2. Proof of Lemma 2.2

*Proof of Lemma 2.2.* We prove it via contradiction. If no such arm exists, meaning that for any arm $a$, there exists an arm $b$ such that $f^\star(x, a, b) < 0$. Then we can find a sequence of arms $(a_1, a_2, \ldots, a_k)$ such that $f^\star(x, a_i, a_{i+1}) < 0$ for any $i = 1, \ldots, k-1$ and $f^\star(x, a_k, a_1) < 0$, which contradicts with the transitivity (Assumption 2.1). $\qquad\square$

### A.3. Proof of Theorem 3.4

We begin by showing the worst-case regret upper bound.

**Lemma A.9** (Worst-case regret upper bound). *For Algorithm 1, assume $f^\star \in \mathcal{F}_t$ for all $t \in [T]$. Then, we have*

$$\mathrm{Regret}_T^{\mathrm{CB}} \leq 68\sqrt{AT\beta} \cdot \log(4\delta^{-1})$$

*with probability at least $1 - \delta$.*

*Proof of Lemma A.9.* We recall that the regret is defined as

$$\mathrm{Regret}_T^{\mathrm{CB}} = \sum_{t=1}^{T} \left( f^\star(x_t, \pi_{f^\star}(x_t), a_t) + f^\star(x_t, \pi_{f^\star}(x_t), b_t) \right).$$

Since $a_t$ and $b_t$ are always drawn independently from the same distribution in Algorithm 1, we only need to consider the regret of the $a_t$ part in the following proof for brevity — multiplying the result by two would yield the overall regret.

We first observe the definition of $\lambda_t$ in Algorithm 1: the left term $\sum_{s=1}^{t-1} Z_s w_s$ in the indicator is non-decreasing in $t$ while the right term remains constant. This means that there exists a particular time step $t' \in [T]$ dividing the time horizon into two phases: $\lambda_t = 0$ for all $t \leq t'$ and $\lambda_t = 1$ for all $t > t'$. Now, we proceed to examine these two phases individually.

For all rounds before $t'$, we can compute the expected partial regret as

$$\sum_{t=1}^{t'} \mathbb{E}_{a \sim p_t} \left[ f^\star(x_t, \pi_{f^\star}(x_t), a) \right] = \sum_{t=1}^{t'} Z_t \mathbb{E}_{a \sim p_t} \left[ f^\star(x_t, \pi_{f^\star}(x_t), a) \right] \leq \sum_{t=1}^{t'} Z_t w_t \leq \sqrt{AT\beta}, \tag{8}$$

where the equality holds since we have $\mathcal{A}_t = \{\pi_{f^\star}(x_t)\}$ whenever $Z_t = 0$ under the condition that $f^\star \in \mathcal{F}_t$. The first inequality is Lemma A.2, and the second inequality holds by the definition of $\lambda_t$ and the condition that $\lambda_t = 0$.

On the other hand, for all rounds after $t'$, we have

$$\sum_{t=t'+1}^{T} \mathbb{E}_{a \sim p_t} \left[ f^\star(x_t, \pi_{f^\star}(x_t), a) \right]$$

$$= \sum_{t=t'+1}^{T} Z_t \mathbb{E}_{a \sim p_t} \left[ f^\star(x_t, \pi_{f^\star}(x_t), a) \right]$$

$$\leq \sum_{t=t'+1}^{T} Z_t \left( \frac{5A}{\gamma_t} + \frac{\gamma_t}{4} \mathbb{E}_{a,b \sim p_t} \left[ \left( f^\star(x_t, a, b) - f_t(x_t, a, b) \right)^2 \right] \right)$$

$$= \sum_{t=t'+1}^{T} Z_t \left( \frac{5A}{\sqrt{AT/\beta}} + \frac{\sqrt{AT/\beta}}{4} \mathbb{E}_{a,b \sim p_t} \left[ \left( f^\star(x_t, a, b) - f_t(x_t, a, b) \right)^2 \right] \right)$$

$$\leq 5\sqrt{AT\beta} + \frac{\sqrt{AT/\beta}}{4} \sum_{t=t'+1}^{T} Z_t \mathbb{E}_{a,b \sim p_t} \left[ \left( f^\star(x_t, a, b) - f_t(x_t, a, b) \right)^2 \right]$$

$$\leq 5\sqrt{AT\beta} + \frac{\sqrt{AT/\beta}}{2} \sum_{t=t'+1}^{T} Z_t \left( f^\star(x_t, a_t, b_t) - f_t(x_t, a_t, b_t) \right)^2 + 8\sqrt{AT/\beta} \cdot \log(4\delta^{-1})$$

$$\leq 5\sqrt{AT\beta} + \frac{\sqrt{AT\beta}}{2} + 8\sqrt{AT/\beta} \cdot \log(4\delta^{-1}). \tag{9}$$

where the first inequality holds by Lemma A.5 (or Lemma A.6 for specific function classes), the second equality is by the definition of $\gamma_t$, the third inequality is by Lemma B.3, and the fourth inequality holds by Lemma A.3.

Putting the two parts, (8) and (9), all together, we arrive at

$$\sum_{t=1}^{T} \mathbb{E}_{a \sim p_t} \left[ f^\star(x_t, \pi_{f^\star}(x_t), a) \right] \leq 7\sqrt{AT\beta} + 8\sqrt{AT/\beta} \cdot \log(4\delta^{-1}) \leq 15\sqrt{AT\beta} \cdot \log(4\delta^{-1}).$$

Now we apply Lemma B.3 again. The following holds with probability at least $1 - \delta/2$,

$$\sum_{t=1}^{T} f^{\star}(x_t, \pi_{f^{\star}}(x_t), a_t) \leq 2 \sum_{t=1}^{T} \mathop{\mathbb{E}}_{a \sim p_t} \left[ f^{\star}(x_t, \pi_{f^{\star}}(x_t), a) \right] + 4 \log(4\delta^{-1}) \leq 34\sqrt{AT\beta} \cdot \log(4\delta^{-1}).$$

The above concludes the regret of the $a_t$ part. The regret of the $b_t$ can be shown in the same way. Adding them together, we conclude that

$$\mathrm{Regret}_T^{\mathrm{CB}} = \sum_{t=1}^{T} \left( f^{\star}(x_t, \pi_{f^{\star}}(x_t), a_t) + f^{\star}(x_t, \pi_{f^{\star}}(x_t), b_t) \right) \leq 68\sqrt{AT\beta} \cdot \log(4\delta^{-1}).$$

$\square$

**Lemma A.10** (Instance-dependent regret upper bound). *For Algorithm 1, assume $f^{\star} \in \mathcal{F}_t$ for all $t \in [T]$. Then, we have*

$$\mathrm{Regret}_T^{\mathrm{CB}} \leq 3808 A^2 \beta^2 \cdot \frac{\dim_E(\mathcal{F}, \Delta)}{\Delta} \cdot \log^2(4/(\delta\Delta))$$

*with probability at least $1 - \delta$.*

*Proof of Lemma A.10.* We consider two cases. First, when

$$56 A^2 \beta \cdot \frac{\dim_E(\mathcal{F}, \Delta)}{\Delta} \cdot \log(2/(\delta\Delta)) < \sqrt{AT/\beta}, \tag{10}$$

we invoke Lemma A.8 and get that $\lambda_t = 0$ for all $t \in [T]$. Hence, we have

$$\begin{aligned}
\mathrm{Regret}_T^{\mathrm{CB}} &= \sum_{t=1}^{T} \left( f^{\star}(x_t, \pi_{f^{\star}}(x_t), a_t) + f^{\star}(x_t, \pi_{f^{\star}}(x_t), b_t) \right) \\
&\leq 2 \sum_{t=1}^{T} Z_t w_t \\
&\leq 112 A^2 \beta \cdot \frac{\dim_E(\mathcal{F}, \Delta)}{\Delta} \cdot \log(2/(\delta\Delta)) \\
&\leq 3808 A^2 \beta^2 \cdot \frac{\dim_E(\mathcal{F}, \Delta)}{\Delta} \cdot \log^2(4/(\delta\Delta))
\end{aligned}$$

where the first inequality is by Lemma A.2 and the fact that we incur no regret when $Z_t = 0$ since $f^{\star} \in \mathcal{F}_t$. The second inequality is by Lemma A.7.

On the other hand, when the contrary of (10) holds, i.e.,

$$56 A^2 \beta \cdot \frac{\dim_E(\mathcal{F}, \Delta)}{\Delta} \cdot \log(2/(\delta\Delta)) \geq \sqrt{AT/\beta}, \tag{11}$$

applying Lemma A.9, we have

$$\begin{aligned}
\mathrm{Regret}_T^{\mathrm{CB}} &\leq 68\sqrt{AT\beta} \cdot \log(4\delta^{-1}) \\
&= 68\beta \cdot \log(4\delta^{-1}) \cdot \sqrt{AT/\beta} \\
&\leq 68\beta \cdot \log(4\delta^{-1}) \cdot 56 A^2 \beta \cdot \frac{\dim_E(\mathcal{F}, \Delta)}{\Delta} \cdot \log(2/(\delta\Delta)) \\
&\leq 3808 A^2 \beta^2 \cdot \frac{\dim_E(\mathcal{F}, \Delta)}{\Delta} \cdot \log^2(4/(\delta\Delta))
\end{aligned}$$

where we apply the condition (11) in the second inequality. $\square$

**Lemma A.11** (Query complexity). *For Algorithm 1, assume $f^\star \in \mathcal{F}_t$ for all $t \in [T]$. Then, we have*

$$\text{Queries}_T^{\text{CB}} \le \min\left\{ T,\ 3136 A^3 \beta^3 \frac{\dim_E^2(\mathcal{F}, \Delta)}{\Delta^2} \cdot \log^2(2/(\delta\Delta)) \right\}$$

*with probability at least $1 - \delta$.*

*Proof of Lemma A.11.* We consider two cases. First, when

$$56 A^2 \beta \cdot \frac{\dim_E(\mathcal{F}, \Delta)}{\Delta} \cdot \log(2/(\delta\Delta)) < \sqrt{AT/\beta} \tag{12}$$

we can invoke Lemma A.8 and get that $\lambda_t = 0$ for all $t \in [T]$. Hence,

$$\begin{aligned}
\text{Queries}_T^{\text{CB}} &= \sum_{t=1}^T Z_t \\
&= \sum_{t=1}^T Z_t \mathbb{1}\{w_t \ge \Delta\} \\
&= \sum_{t=1}^T Z_t \sup_{a,b \in \mathcal{A}_t} \mathbb{1}\left\{ \sup_{f,f' \in \mathcal{F}_t} f(x_t, a, b) - f'(x_t, a, b) \ge \Delta \right\} \\
&\le \sum_{t=1}^T Z_t \sum_{a,b} \mathbb{1}\left\{ \sup_{f,f' \in \mathcal{F}_t} f(x_t, a, b) - f'(x_t, a, b) \ge \Delta \right\} \\
&\le A^2 \underbrace{\sum_{t=1}^T Z_t \mathop{\mathbb{E}}_{a,b \sim p_t} \mathbb{1}\left\{ \sup_{f,f' \in \mathcal{F}_t} f(x_t, a, b) - f'(x_t, a, b) \ge \Delta \right\}}_{(*)}
\end{aligned}$$

where the second equality is by Lemma A.1, the second inequality holds as $p_t(a)$ is uniform for any $a, b$ when $\lambda_t = 0$. We apply Lemma B.3 and Lemma A.4 to $(*)$ and obtain

$$\begin{aligned}
(*) &\le 2 \sum_{t=1}^T Z_t \mathbb{1}\left\{ \sup_{f,f' \in \mathcal{F}_t} f(x_t, a_t, b_t) - f'(x_t, a_t, b_t) \ge \Delta \right\} + 8 \log(\delta^{-1}) \\
&\le 2\left( \frac{4\beta}{\Delta^2} + 1 \right) \dim_E(\mathcal{F}; \Delta) + 8 \log(\delta^{-1}) \\
&\le \frac{10\beta}{\Delta^2} \cdot \dim_E(\mathcal{F}; \Delta) + 8 \log(\delta^{-1}).
\end{aligned}$$

Plugging this back, we obtain

$$\begin{aligned}
\text{Queries}_T^{\text{CB}} &\le \frac{10 A^2 \beta}{\Delta^2} \cdot \dim_E(\mathcal{F}; \Delta) + 8 A^2 \log(\delta^{-1}) \\
&\le 3136 A^3 \beta^3 \frac{\dim_E^2(\mathcal{F}, \Delta)}{\Delta^2} \cdot \log^2(2/(\delta\Delta)).
\end{aligned}$$

On the other hand, when the contrary of (12) holds, i.e.,

$$56 A^2 \beta \cdot \frac{\dim_E(\mathcal{F}, \Delta)}{\Delta} \cdot \log(2/(\delta\Delta)) \ge \sqrt{AT/\beta}.$$

Squaring both sides, we obtain

$$3136 A^4 \beta^2 \frac{\dim_E^2(\mathcal{F}, \Delta)}{\Delta^2} \cdot \log^2(2/(\delta\Delta)) \ge AT/\beta$$

which leads to

$$T \leq 3136 A^3 \beta^3 \frac{\dim_E^2(\mathcal{F}, \Delta)}{\Delta^2} \cdot \log^2(2/(\delta\Delta)).$$

We note that we always have $\text{Queries}_T^{\text{CB}} \leq T$, and thus,

$$\text{Queries}_T^{\text{CB}} \leq T \leq 3136 A^3 \beta^3 \frac{\dim_E^2(\mathcal{F}, \Delta)}{\Delta^2} \cdot \log^2(2/(\delta\Delta)).$$

Hence, we complete the proof. $\qquad\square$

Having established the aforementioned lemmas, we are now able to advance towards the proof of Theorem 1.

*Proof of Theorem 3.4.* By Lemma A.3 and the construction of version spaces $\mathcal{F}_t$ in Algorithm 1, we have $f^\star \in \mathcal{F}_t$ for all $t \in [T]$ with probability at least $1 - \delta$. Then, the rest of the proof follows from Lemmas A.9 to A.11. $\qquad\square$

## A.4. Proof of Theorem 3.5

In this section, we will prove the following theorem, which is stronger than Theorem 3.5.

**Theorem A.12** (Lower bounds). *The following two claims hold:*

*(1) for any algorithm, there exists an instance that leads to* $\text{Regret}_T^{\text{CB}} \geq \Omega(\sqrt{AT})$;

*(2) for any algorithm achieving a worse-case expected regret upper bound in the form of* $\mathbb{E}[\text{Regret}_T^{\text{CB}}] \leq O(\sqrt{A} \cdot T^{1-\beta})$ *for some* $\beta > 0$, *there exists an instance with gap* $\Delta = \sqrt{A} \cdot T^{-\beta}$ *that results in* $\mathbb{E}[\text{Regret}_T^{\text{CB}}] \geq \Omega(A/\Delta) = \Omega(\sqrt{A} \cdot T^\beta)$ *and* $\mathbb{E}[\text{Queries}_T^{\text{CB}}] \geq \Omega(A/\Delta^2) = \Omega(T^{2\beta})$.

We observe that Theorem 3.5 can be considered as a corollary of the above theorem when setting $\beta = 1/2$.

In what follows, we will first demonstrate lower bounds in the setting of *multi-armed bandits (MAB) with active queries* and subsequently establish a reduction from it to contextual dueling bandits in order to activate these lower bounds. We start by formally define the setting of MAB with active queries below.

**Multi-armed bandits with active queries.** We consider a scenario where there exist $A$ arms. Each arm $a$ is assumed to yield a binary reward (0 or 1), which is sampled from a Bernoulli distribution $\text{Bern}(\bar{r}_a)$, where $\bar{r}_a$ denotes the mean reward associated with arm $a$. The arm with the highest mean reward is denoted by $a^\star := \arg\max_a \bar{r}_a$. Let $\Delta_a := \bar{r}_{a^\star} - \bar{r}_a$ denote the gap of arm $a \in [A]$. The interaction proceeds as follows: at each round $t \in [T]$, we need to pull an arm but can choose whether to receive the reward signal (denote this choice by $Z_t$). The objective is to minimize two quantities: the regret and the number of queries,

$$\text{Regret}_T = \sum_{t=1}^{T} \Delta_{a_t}, \quad \text{Queries}_T = \sum_{t=1}^{T} Z_t. \tag{13}$$

Towards the lower bounds, we will start with a bound on the KL divergence over distributions of runs under two different bandits. This result is a variant of standard results which can be found in many bandit literature (e.g., Lattimore & Szepesvári (2020)).

**Lemma A.13.** *Let $I_1$ and $I_2$ be two instances of MAB. We define $p_1$ and $p_2$ as their respective distributions over the outcomes of all pulled arms and reward signals **when a query is made**. Concretely, $p_1$ and $p_2$ are measuring the probability of outcomes (denoted by $O$) in the following form:*

$$O = (Z_1, a_1, (r_1), \ldots, Z_T, a_T, (r_T))$$

*where the reward $r_t$ is included only when $Z_t = 1$, and we added parentheses above to indicate this point. We denote $\text{Pr}_1$ (resp. $\text{Pr}_2$) as the reward distribution of $I_1$ (resp. $I_2$). We define $\bar{n}_a = \sum_{t=1}^{T} Z_t \mathbb{1}\{a_t = a\}$ as the number of times arm $a$ is pulled when making a query. Then, given any algorithm $\mathbf{A}$, the Kullback–Leibler divergence between $p_1$ and $p_2$ can be decomposed in the following way*

$$\text{KL}(p_1, p_2) = \sum_{a=1}^{A} \mathbb{E}_{p_1}[\bar{n}_a] \cdot \text{KL}\big(\text{Pr}_1(r \mid a), \text{Pr}_2(r \mid a)\big).$$

*Proof of Lemma A.13.* We define the conditional distribution

$$\overline{\Pr}_1(r_t \mid Z_t, a_t) \begin{cases} \Pr_1(r_t \mid a_t) & \text{if } Z_t = 1 \\ 1 & \text{if } Z_t = 0 \end{cases},$$

and similarly for $\overline{\Pr}_2$. Additionally, we denote $\Pr_{\boldsymbol{A}}$ as the probability associated with algorithm $\boldsymbol{A}$. Then, for any outcome $O$, we have

$$p_1(O) = \prod_{t=1}^{T} \Pr_{\boldsymbol{A}} \left( Z_t, a_t \mid Z_1, a_1, (r_1), \ldots, Z_{t-1}, a_{t-1}, (r_{t-1}) \right) \overline{\Pr}_1(r_t \mid Z_t, a_t),$$

and we can write $p_2(O)$ in a similar manner. Hence,

$$\begin{aligned}
\text{KL}(p_1, p_2) &= \underset{O \sim p_1}{\mathbb{E}} \left[ \log \left( \frac{\prod_{t=1}^{T} \Pr_{\boldsymbol{A}} \left( Z_t, a_t \mid Z_1, a_1, (r_1), \ldots, Z_{t-1}, a_{t-1}, (r_{t-1}) \right) \overline{\Pr}_1(r_t \mid Z_t, a_t)}{\prod_{t=1}^{T} \Pr_{\boldsymbol{A}} \left( Z_t, a_t \mid Z_1, a_1, (r_1), \ldots, Z_{t-1}, a_{t-1}, (r_{t-1}) \right) \overline{\Pr}_2(r_t \mid Z_t, a_t)} \right) \right] \\
&= \underset{O \sim p_1}{\mathbb{E}} \left[ \sum_{t=1}^{T} \log \left( \frac{\overline{\Pr}_1(r_t \mid Z_t, a_t)}{\overline{\Pr}_2(r_t \mid Z_t, a_t)} \right) \right] \\
&= \underset{O \sim p_1}{\mathbb{E}} \left[ \sum_{t=1}^{T} Z_t \log \left( \frac{\Pr_1(r_t \mid a_t)}{\Pr_2(r_t \mid a_t)} \right) \right] \\
&= \underset{O \sim p_1}{\mathbb{E}} \left[ \sum_{t=1}^{T} Z_t \underset{r_t \sim \Pr_1(\cdot \mid a_t)}{\mathbb{E}} \left[ \log \left( \frac{\Pr_1(r_t \mid a_t)}{\Pr_2(r_t \mid a_t)} \right) \right] \right] \\
&= \underset{O \sim p_1}{\mathbb{E}} \left[ \sum_{t=1}^{T} Z_t \cdot \text{KL} \left( \Pr_1(\cdot \mid a_t), \Pr_2(\cdot \mid a_t) \right) \right] \\
&= \sum_{a=1}^{A} \underset{O \sim p_1}{\mathbb{E}} \left[ \bar{n}_a \right] \cdot \text{KL} \left( \Pr_1(\cdot \mid a_t), \Pr_2(\cdot \mid a_t) \right)
\end{aligned}$$

where the third equality holds by the definition of $\overline{\Pr}_1$ and $\overline{\Pr}_2$. $\qquad\square$

The following lemma establishes lower bounds for MAB with active queries. It presents a trade-off between the regret and the number of queries.

**Lemma A.14.** *Let $\mathcal{I}$ denote the set of all MAB instances. Assume $\boldsymbol{A}$ is an algorithm that achieves the following worst-case regret upper bound for some $C$ and $\beta$:*

$$\mathbb{E} \left[ \text{Regret}_T \right] \le C T^{1-\beta},$$

*for all $I \in \mathcal{I}$. Then, for any MAB instance $I \in \mathcal{I}$, the regret and the number of queries made by algorithm $\boldsymbol{A}$ are lower bounded:*

$$\mathbb{E} \left[ \text{Regret}_T \right] \ge \sum_{a \ne a^\star} \frac{\zeta}{\Delta_a} \log \left( \frac{\Delta_a}{4CT^{-\beta}} \right), \quad \mathbb{E} \left[ \text{Queries}_T \right] \ge \sum_{a \ne a^\star} \frac{\zeta}{\Delta_a^2} \log \left( \frac{\Delta_a}{4CT^{-\beta}} \right)$$

*where the coefficient $\zeta = \min_a \min\{\bar{r}_a, 1 - \bar{r}_a\}$ depends on the instance $I$.*

*Proof of Lemma A.14.* For any MAB instance $I$ and any arm $a^\dagger$, we define a corresponding MAB instance $I'$ as follows. Denote $\bar{r}$ and $\bar{r}'$ as the mean reward of $I$ and $I'$, respectively. For $I'$, we set the mean reward $\bar{r}'_a = \bar{r}_a$ for any $a \ne a^\dagger$ and $\bar{r}'_{a^\dagger} = \bar{r}_{a^\dagger} + 2\Delta_{a^\dagger}$. Consequently, the optimal arm of $I'$ is $a^\dagger$ with margin $\Delta_{a^\dagger}$. Let $n_a$ denote the number of times that arm $a$ is pulled. We define the event

$$E = \{n_{a^\dagger} > T/2\}.$$

Then, we have

$$\underset{p}{\mathbb{E}} \left[ \text{Regret}_T \right] \ge \frac{T\Delta_{a^\dagger}}{2} \cdot p(E), \quad \underset{p'}{\mathbb{E}} \left[ \text{Regret}_T \right] \ge \frac{T\Delta_{a^\dagger}}{2} \cdot p'(E^{\complement}).$$

Hence,

$$
\begin{aligned}
2CT^{1-\beta} &\geq \mathop{\mathbb{E}}_{p}\left[\mathrm{Regret}_T\right] + \mathop{\mathbb{E}}_{p'}\left[\mathrm{Regret}_T\right] \\
&\geq \frac{T\Delta_{a^\dagger}}{2}\left(p(E) + p'(E^\complement)\right) \\
&= \frac{T\Delta_{a^\dagger}}{2}\left(1 - \left(p'(E) - p(E)\right)\right) \\
&\geq \frac{T\Delta_{a^\dagger}}{2}\left(1 - \mathrm{TV}(p, p')\right) \\
&\geq \frac{T\Delta_{a^\dagger}}{2}\left(1 - \sqrt{1 - \exp\left(-\,\mathrm{KL}(p, p')\right)}\right) \\
&\geq \frac{T\Delta_{a^\dagger}}{2}\exp\left(-\frac{1}{2}\cdot\mathrm{KL}(p, p')\right).
\end{aligned}
$$

By Lemma A.13, we have

$$
\begin{aligned}
\mathrm{KL}(p, p') &= \sum_{a=1}^{A}\mathop{\mathbb{E}}_{p}[\bar{n}_a]\cdot\mathrm{KL}\left(\Pr(r\,|\,a), \Pr'(r\,|\,a)\right) \\
&= \mathop{\mathbb{E}}_{p}[\bar{n}_{a^\dagger}]\cdot\mathrm{KL}\left(\Pr(r\,|\,a^\dagger), \Pr'(r\,|\,a^\dagger)\right) \\
&\leq \mathop{\mathbb{E}}_{p}[\bar{n}_{a^\dagger}]\cdot\Delta_{a^\dagger}^2\cdot 2/\zeta
\end{aligned}
$$

where the last inequality is by Lemma B.5. Putting the above two inequality together, we arrive at

$$
\mathop{\mathbb{E}}_{p}[\bar{n}_{a^\dagger}] \geq \frac{\zeta}{\Delta_{a^\dagger}^2}\log\left(\frac{\Delta_{a^\dagger}}{4CT^{-\beta}}\right).
$$

This establishes a query lower bound for arm $a^\dagger$. Consequently, we have

$$
\mathbb{E}[\mathrm{Regret}_T] \geq \sum_{a\neq a^\star}\mathop{\mathbb{E}}_{p}[\bar{n}_a]\cdot\Delta_a \geq \sum_{a\neq a^\star}\frac{\zeta}{\Delta_a}\log\left(\frac{\Delta_a}{4CT^{-\beta}}\right),
$$

and similarly,

$$
\mathbb{E}[\mathrm{Queries}_T] \geq \sum_{a\neq a^\star}\mathop{\mathbb{E}}_{p}[\bar{n}_a] \geq \sum_{a\neq a^\star}\frac{\zeta}{\Delta_a^2}\log\left(\frac{\Delta_a}{4CT^{-\beta}}\right).
$$

$\square$

Now we can proceed with the proof of Theorem A.12.

*Proof of Theorem A.12.* We will show a reduction from the the multi-armed bandits with active queries to the contextual dueling bandits in order to apply lower bounds from the MAB literature and the ones established above.

**Reduction.** Since we focus on the multi-armed bandit where no context is involved, we just ignore the notation of context everywhere for brevity. For the dueling setting, given two arms $a$ and $b$, we define $f^\star(a, b) = \bar{r}_a - \bar{r}_b$ and let the preference-based feedback be obtained in the following process:

1. We pull both arm once, obtaining reward $y_a, y_b \in \{0, 1\}$.

2. If $y_a > y_b$, we return 1. If $y_a < y_b$, we return $-1$. Otherwise (i.e., $y_a = y_b = 1$ or 0), we return either $-1$ or 1 with equal probability, $1/2$.

Then we can verify that the probability of returning 1 is $(\bar{r}_a - \bar{r}_b + 1)/2$. So we can just specify the link function to be $\phi(d) = (d+1)/2$. As we verified earlier, $\Phi = \int \phi$ is strongly convex (Example 2). Moreover, if we define the gap of an MAB as $\bar{\Delta} := \min_{a \neq a^\star}(\bar{r}_{a^\star} - \bar{r}_a)$ where $a^\star := \arg\max_i \bar{r}_i$, then we have $\bar{\Delta} = \Delta$ in this reduction. We futher note that the regret of MAB is

$$\sum_{t=1}^{T}(\bar{r}_{a^\star} - \bar{r}_{a_t}) + \sum_{t=1}^{T}(\bar{r}_{a^\star} - \bar{r}_{b_t}),$$

which, by our definition of $f^\star$, is equivalent to the preference-based regret. The number of queries is clearly equivalent as well.

Now, we are ready to prove the two claims in our statement.

**Proof of the first claim.** We refer the reader to Lattimore & Szepesvári (2020, Theorem 15.2) for a proof of the minimax regret lower bound of $\Omega(\sqrt{AT})$ for the MAB. Through the reduction outlined above, that lower bound naturally extends to the dueling bandits setting, yielding $\mathrm{Regret}_T^{\mathrm{CB}} \geq \Omega(\sqrt{AT})$.

**Proof of the second claim.** We choose an arbitrary MAB for which $\zeta = \min_a \min\{\bar{r}_a, 1 - \bar{r}_a\} > 0.2$ and the gaps of all arms are equal to $\Delta$. Invoking Lemma A.14, we have

$$\mathbb{E}\left[\mathrm{Regret}_T\right] \geq \frac{0.2(A-1)}{\Delta} \log\left(\frac{\Delta}{4CT^{-\beta}}\right) \geq \Omega\left(\frac{A}{\Delta}\right),$$

$$\mathbb{E}\left[\mathrm{Queries}_T\right] \geq \frac{0.2(A-1)}{\Delta^2} \log\left(\frac{\Delta}{4CT^{-\beta}}\right) \geq \Omega\left(\frac{A}{\Delta^2}\right).$$

We further choose $\Delta = 40CT^{-\beta}$ and $C = \sqrt{A}$, leading to

$$\mathbb{E}\left[\mathrm{Regret}_T\right] \geq \frac{0.2(A-1)}{40\sqrt{A}} \cdot T^{\beta} = \Omega\left(\sqrt{A} \cdot T^{\beta}\right),$$

$$\mathbb{E}\left[\mathrm{Queries}_T\right] \geq \frac{0.2(A-1)}{1600A} \cdot T^{2\beta} = \Omega\left(T^{2\beta}\right).$$

Via the reduction we have shown above, these lower bounds naturally extend to the contextual dueling bandit setting, thereby completing the proof. $\square$

### A.4.1. ALTERNATIVE LOWER BOUNDS CONDITIONING ON THE LIMIT OF REGRET

In this section, we establish an analogue of Theorem A.12 but under a different condition. We first introduce the concept of *diminishing regret*.

**Definition A.15.** We say that an algorithm guarantees a diminishing regret if for all contextual dueling bandit instances and $p > 0$, it holds that

$$\lim_{T \to \infty} \frac{\mathbb{E}[\mathrm{Regret}_T^{\mathrm{CB}}]}{T^p} = 0.$$

The lower bounds under the assumption of diminishing regret guarantees are stated as follows.

**Theorem A.16** (Lower bounds)**.** *The following two claims hold:*

*(1) for any algorithm, there exists an instance that leads to $\mathrm{Regret}_T^{\mathrm{CB}} \geq \Omega(\sqrt{AT})$;*

*(2) for any gap $\Delta$ and any algorithm achieving diminishing regret, there exists an instance with gap $\Delta$ that results in $\mathbb{E}[\mathrm{Regret}_T^{\mathrm{CB}}] \geq \Omega(A/\Delta)$ and $\mathbb{E}[\mathrm{Queries}_T^{\mathrm{CB}}] \geq \Omega(A/\Delta^2)$ for sufficiently large $T$.*

We should highlight that the condition of diminishing regret (Theorem A.16) and the worst-case regret upper bounds (Theorems 3.5 and A.12) are not comparable in general. However, Theorem A.16 is also applicable to our algorithm (Algorithm 1) since our algorithm possesses an instance-dependent regret upper bound that is clearly diminishing.

To prove Theorem A.16, we first show the following lemma, which is a variant of Lemma A.14.

**Lemma A.17.** *Let $\mathcal{I}$ denote the set of all MAB instances. Assume $\mathbf{A}$ is an algorithm that achieves diminishing regret for all MAB instances in $\mathcal{I}$, i.e., for any $I \in \mathcal{I}$ and $p > 0$, it holds that*

$$\lim_{T \to \infty} \frac{\mathbb{E}[\mathrm{Regret}_T]}{T^p} = 0.$$

*Then, for any MAB instance $I \in \mathcal{I}$, the regret and the number of queries made by algorithm $\mathbf{A}$ are lower bounded in the following manner:*

$$\liminf_{T \to \infty} \frac{\mathbb{E}\left[\mathrm{Regret}_T\right]}{\log T} \geq \sum_{a \neq a^\star} \frac{\zeta}{\Delta_a}, \quad \liminf_{T \to \infty} \frac{\mathbb{E}\left[\mathrm{Queries}_T\right]}{\log T} \geq \sum_{a \neq a^\star} \frac{\zeta}{\Delta_a^2}$$

*where the coefficient $\zeta := \min_a \min\{\bar{r}_a, 1 - \bar{r}_a\}$ depends on the instance $I$. Recall that $\mathrm{Regret}_T$ and $\mathrm{Queries}_T$ are defined in* (13).

*Proof of Lemma A.17.* The proof is similar to Lemma A.14. For any MAB instance $I \in \mathcal{I}$ and any arm $a^\dagger$, we define a corresponding MAB instance $I'$ as follows. Denote $\bar{r}$ and $\bar{r}'$ as the mean reward of $I$ and $I'$, respectively. For $I'$, we set the mean reward $\bar{r}'_a = \bar{r}_a$ for any $a \neq a^\dagger$ and $\bar{r}'_{a^\dagger} = \bar{r}_{a^\dagger} + 2\Delta_{a^\dagger}$. Consequently, the optimal arm of $I'$ is $a^\dagger$ with margin $\Delta_{a^\dagger}$. Let $n_a$ denote the number of times that arm $a$ is pulled. We define the event

$$E = \{n_{a^\dagger} > T/2\}.$$

Let $p$ and $p'$ denote the probability of $I$ and $I'$, respectively. Then, we have

$$\mathbb{E}_p\left[\mathrm{Regret}_T\right] \geq \frac{T\Delta_{a^\dagger}}{2} \cdot p(E), \quad \mathbb{E}_{p'}\left[\mathrm{Regret}_T\right] \geq \frac{T\Delta_{a^\dagger}}{2} \cdot p'(E^{\complement})$$

where $E^{\complement}$ means the complement of event $E$. Hence,

$$
\begin{aligned}
\mathbb{E}_p\left[\mathrm{Regret}_T\right] + \mathbb{E}_{p'}\left[\mathrm{Regret}_T\right] &\geq \frac{T\Delta_{a^\dagger}}{2}\left(p(E) + p'(E^{\complement})\right) \\
&= \frac{T\Delta_{a^\dagger}}{2}\left(1 - \left(p'(E) - p(E)\right)\right) \\
&\geq \frac{T\Delta_{a^\dagger}}{2}\left(1 - \mathrm{TV}\left(p, p'\right)\right) \\
&\geq \frac{T\Delta_{a^\dagger}}{2}\left(1 - \sqrt{1 - \exp\left(-\mathrm{KL}(p, p')\right)}\right) \\
&\geq \frac{T\Delta_{a^\dagger}}{2}\exp\left(-\frac{1}{2} \cdot \mathrm{KL}(p, p')\right).
\end{aligned}
$$

Here TV denotes the total variation distance. By Lemma A.13, we have

$$
\begin{aligned}
\mathrm{KL}(p, p') &= \sum_{a=1}^{A} \mathbb{E}_p[\bar{n}_a] \cdot \mathrm{KL}\left(\mathrm{Pr}(r \mid a), \mathrm{Pr}'(r \mid a)\right) \\
&= \mathbb{E}_p[\bar{n}_{a^\dagger}] \cdot \mathrm{KL}\left(\mathrm{Pr}(r \mid a^\dagger), \mathrm{Pr}'(r \mid a^\dagger)\right) \\
&\leq \mathbb{E}_p[\bar{n}_{a^\dagger}] \cdot \Delta_{a^\dagger}^2 \cdot 2/\zeta
\end{aligned}
$$

where the last inequality is by Lemma B.5. Putting it all together, we arrive at

$$\mathbb{E}_p[\bar{n}_{a^\dagger}] \geq \frac{\zeta}{\Delta_{a^\dagger}^2} \log\left(\frac{T\Delta_{a^\dagger}}{2\left(\mathbb{E}_p\left[\mathrm{Regret}_T\right] + \mathbb{E}_{p'}\left[\mathrm{Regret}_T\right]\right)}\right).$$

Taking the limit on both sides yields

$$
\liminf_{T\to\infty} \frac{\mathbb{E}_p[\bar{n}_{a^\dagger}]}{\log T} \geq \liminf_{T\to\infty} \frac{\zeta}{\Delta_{a^\dagger}^2} \cdot \frac{\log\left(\frac{T\Delta_{a^\dagger}}{2\left(\mathbb{E}_p\left[\mathrm{Regret}_T\right]+\mathbb{E}_{p'}\left[\mathrm{Regret}_T\right]\right)}\right)}{\log T}
$$

$$
= \liminf_{T\to\infty} \frac{\zeta}{\Delta_{a^\dagger}^2} \cdot \left(1 + \underbrace{\frac{\log(\Delta_{a^\dagger}/2)}{\log T}}_{(i)} - \underbrace{\frac{\log\left(\mathbb{E}_p\left[\mathrm{Regret}_T\right]+\mathbb{E}_{p'}\left[\mathrm{Regret}_T\right]\right)}{\log T}}_{(ii)}\right).
$$

Here the limit of (i) is clearly 0. For the limit of (ii), we note that by the definition of diminishing regret, for any $C > 0$, there exists a $T'$ such that $\mathbb{E}[\mathrm{Regret}_T]/T^p \leq C$ for any $T > T'$. This implies

$$
\frac{\log\left(\mathbb{E}_p\left[\mathrm{Regret}_T\right]+\mathbb{E}_{p'}\left[\mathrm{Regret}_T\right]\right)}{\log T} \leq \frac{\log\left(2CT^p\right)}{\log T} = \frac{\log(2C)}{\log T} + p
$$

for any $p > 0$. Therefore, the limit of (ii) is also 0. Plugging these back, we obtain

$$
\liminf_{T\to\infty} \frac{\mathbb{E}_p[\bar{n}_{a^\dagger}]}{\log T} \geq \frac{\zeta}{\Delta_{a^\dagger}^2}.
$$

This establishes a query lower bound for arm $a^\dagger$. Consequently, we have

$$
\liminf_{T\to\infty} \frac{\mathbb{E}[\mathrm{Regret}_T]}{\log T} \geq \liminf_{T\to\infty} \sum_{a\neq a^\star} \frac{\mathbb{E}_p[\bar{n}_a] \cdot \Delta_a}{\log T} \geq \sum_{a\neq a^\star} \frac{\zeta}{\Delta_a},
$$

and similarly,

$$
\liminf_{T\to\infty} \frac{\mathbb{E}[\mathrm{Queries}_T]}{\log T} \geq \liminf_{T\to\infty} \sum_{a\neq a^\star} \frac{\mathbb{E}_p[\bar{n}_a]}{\log T} \geq \sum_{a\neq a^\star} \frac{\zeta}{\Delta_a^2}.
$$

$\square$

Now, we proceed with the proof of Theorem A.16.

*Proof of Theorem A.16.* The proof of the first claim is the same as Theorem A.12, so we will omit it here. Let us now focus on the proof of the second claim. By Lemma A.17, for any algorithm achieving diminishing regret, the following is true for any MAB instance:

$$
\liminf_{T\to\infty} \frac{\mathbb{E}\left[\mathrm{Regret}_T\right]}{\log T} \geq \sum_{a\neq a^\star} \frac{\zeta}{\Delta_a}, \quad \liminf_{T\to\infty} \frac{\mathbb{E}\left[\mathrm{Queries}_T\right]}{\log T} \geq \sum_{a\neq a^\star} \frac{\zeta}{\Delta_a^2}.
$$

We choose an arbitrary MAB for which $\zeta \geq 0.2$ and the gaps of all suboptimal arms are equal to $\Delta$. Then, for this instance, we have

$$
\liminf_{T\to\infty} \frac{\mathbb{E}\left[\mathrm{Regret}_T\right]}{\log T} \geq \frac{0.2(A-1)}{\Delta}, \quad \liminf_{T\to\infty} \frac{\mathbb{E}\left[\mathrm{Queries}_T\right]}{\log T} \geq \frac{0.2(A-1)}{\Delta^2}.
$$

By the definition of limit, when $T$ is large enough (exceeding a certain threshold), we have

$$
\frac{\mathbb{E}\left[\mathrm{Regret}_T\right]}{\log T} \geq \frac{0.1(A-1)}{\Delta}, \quad \frac{\mathbb{E}\left[\mathrm{Queries}_T\right]}{\log T} \geq \frac{0.1(A-1)}{\Delta^2}.
$$

Via the reduction we have shown in the proof of Theorem A.12, these lower bounds naturally extend to the contextual dueling bandit setting, thereby completing the proof. $\square$

## A.5. Proof of Theorem 3.6

*Proof of Theorem 3.6.* We establish the bounds for regret and the number of queries, consecutively. First, we set an arbitrary gap threshold $\epsilon > 0$. Since our algorithm is independent of $\epsilon$, we can later choose any $\epsilon$ that minimizes the upper bounds.

**Proof of regret.** We start with the regret upper bound. By definition, we have

$$\text{Regret}_T^{\text{CB}} = \sum_{t=1}^{T} \left( f^\star(x_t, \pi_{f^\star}(x_t), a_t) + f^\star(x_t, \pi_{f^\star}(x_t), b_t) \right).$$

Since $a_t$ and $b_t$ are always drawn independently from the same distribution in Algorithm 1, we only need to consider the regret of the $a_t$ part in the following proof for brevity — multiplying the result by two would yield the overall regret.

The worst-case regret upper bound presented in Lemma A.9 doesn't reply on the gap assumption and thus remains applicable in this setting. Hence, we only need to prove the instance-dependent regret upper bound. To that end, we first need an analogue of Lemma A.8.

**Lemma A.18.** *Fix any $\epsilon > 0$. Whenever*

$$2T_\epsilon + 56A^2\beta \cdot \frac{\dim_E(\mathcal{F}, \epsilon)}{\epsilon} \cdot \log(2/(\delta\epsilon)) < \sqrt{AT/\beta},$$

*we have $\lambda_1 = \lambda_2 = \cdots = \lambda_T = 0$ with probability at least $1 - \delta$.*

*Proof of Lemma A.18.* The proof is similar to Lemma A.8 and is via contradiction. Assume the inequality holds but there exists $t'$ for which $\lambda_{t'} = 1$. Without loss of generality, we assume that $\lambda_t = 0$ for all $t < t'$, namely that $t'$ is the first time that $\lambda_t$ is 1. Then by definition of $\lambda_{t'}$, we have

$$\sum_{s=1}^{t'-1} Z_s w_s \geq \sqrt{AT/\beta}.$$

On the other hand, we have

$$\sum_{s=1}^{t'-1} Z_s w_s = \sum_{s=1}^{t'-1} \mathbb{1}\{\text{Gap}(x_t) \leq \epsilon\} Z_s w_s + \sum_{s=1}^{t'-1} \mathbb{1}\{\text{Gap}(x_t) > \epsilon\} Z_s w_s$$
$$\leq 2T_\epsilon + 56A^2\beta \cdot \frac{\dim_E(\mathcal{F}, \epsilon)}{\epsilon} \cdot \log(2/(\delta\epsilon))$$

where the inequality is by Lemma A.7. The above two inequalities contradicts with the conditions. $\square$

Towards an instance-dependent regret upper bound, we adapt the proof of Lemma A.10 to this setting. We consider two cases. First, when

$$2T_\epsilon + 56A^2\beta \cdot \frac{\dim_E(\mathcal{F}, \epsilon)}{\epsilon} \cdot \log(2/(\delta\epsilon)) < \sqrt{AT/\beta}, \tag{14}$$

we invoke Lemma A.18 and get that $\lambda_t = 0$ for all $t \in [T]$. Hence, we have

$$\text{Regret}_T^{\text{CB}} = \sum_{t=1}^{T} \left( f^\star(x_t, \pi_{f^\star}(x_t), a_t) + f^\star(x_t, \pi_{f^\star}(x_t), b_t) \right)$$
$$\leq 2\sum_{t=1}^{T} \mathbb{1}\{\text{Gap}(x_t) \leq \epsilon\} Z_t w_t + 2\sum_{t=1}^{T} \mathbb{1}\{\text{Gap}(x_t) > \epsilon\} Z_t w_t$$
$$\leq 4T_\epsilon + 112A^2\beta \cdot \frac{\dim_E(\mathcal{F}, \epsilon)}{\epsilon} \cdot \log(2/(\delta\epsilon))$$
$$\leq 136\beta \cdot \log(4\delta^{-1}) \cdot T_\epsilon + 3808A^2\beta^2 \cdot \frac{\dim_E(\mathcal{F}, \epsilon)}{\epsilon} \cdot \log^2(4/(\delta\epsilon))$$

where the first inequality is by Lemma A.2 and the fact that we incur no regret when $Z_t = 0$ since $f^\star \in \mathcal{F}_t$. The second inequality is by Lemma A.7.

On the other hand, when the contrary of (14) holds, i.e.,

$$2T_\epsilon + 56A^2\beta \cdot \frac{\dim_E(\mathcal{F},\epsilon)}{\epsilon} \cdot \log(2/(\delta\epsilon)) \geq \sqrt{AT/\beta}, \tag{15}$$

applying Lemma A.9, we have

$$\begin{aligned}
\mathrm{Regret}_T^{\mathrm{CB}} \leq &68\sqrt{AT\beta} \cdot \log(4\delta^{-1}) \\
= &68\beta \cdot \log(4\delta^{-1}) \cdot \sqrt{AT/\beta} \\
\leq &68\beta \cdot \log(4\delta^{-1}) \cdot \left(2T_\epsilon + 56A^2\beta \cdot \frac{\dim_E(\mathcal{F},\epsilon)}{\epsilon} \cdot \log(2/(\delta\epsilon))\right) \\
\leq &136\beta \cdot \log(4\delta^{-1}) \cdot T_\epsilon + 3808A^2\beta^2 \cdot \frac{\dim_E(\mathcal{F},\epsilon)}{\epsilon} \cdot \log^2(4/(\delta\epsilon))
\end{aligned}$$

where we apply the condition (15) in the second inequality.

**Proof of the number of queries.** To show an upper bound for the number of queries, we also consider two cases. First, when

$$2T_\epsilon + 56A^2\beta \cdot \frac{\dim_E(\mathcal{F},\epsilon)}{\epsilon} \cdot \log(2/(\delta\epsilon)) < \sqrt{AT/\beta}, \tag{16}$$

we can invoke Lemma A.18 and get that $\lambda_t = 0$ for all $t \in [T]$. Hence, similar to the proof of Lemma A.11, we have

$$\begin{aligned}
\mathrm{Queries}_T^{\mathrm{CB}} = &\sum_{t=1}^T Z_t \\
= &\sum_{t=1}^T Z_t \mathbb{1}\{\mathrm{Gap}(x_t) < \epsilon\} + \sum_{t=1}^T Z_t \mathbb{1}\{\mathrm{Gap}(x_t) \geq \epsilon\} \\
= &T_\epsilon + \sum_{t=1}^T Z_t \sup_{a,b\in\mathcal{A}_t} \mathbb{1}\left\{\sup_{f,f'\in\mathcal{F}_t} f(x_t,a,b) - f'(x_t,a,b) \geq \epsilon\right\} \\
\leq &T_\epsilon + \sum_{t=1}^T Z_t \sum_{a,b} \mathbb{1}\left\{\sup_{f,f'\in\mathcal{F}_t} f(x_t,a,b) - f'(x_t,a,b) \geq \epsilon\right\} \\
\leq &T_\epsilon + A^2 \underbrace{\sum_{t=1}^T Z_t \mathbb{E}_{a,b\sim p_t} \mathbb{1}\left\{\sup_{f,f'\in\mathcal{F}_t} f(x_t,a,b) - f'(x_t,a,b) \geq \epsilon\right\}}_{(*)}
\end{aligned}$$

where the second inequality holds as $p_t(a)$ is uniform for any $a,b$ when $\lambda_t = 0$. We apply Lemma B.3 and Lemma A.4 to $(*)$ and obtain

$$\begin{aligned}
(*) \leq &2\sum_{t=1}^T Z_t \mathbb{1}\left\{\sup_{f,f'\in\mathcal{F}_t} f(x_t,a_t,b_t) - f'(x_t,a_t,b_t) \geq \epsilon\right\} + 8\log(\delta^{-1}) \\
\leq &2\left(\frac{4\beta}{\epsilon^2} + 1\right)\dim_E(\mathcal{F};\epsilon) + 8\log(\delta^{-1}) \\
\leq &\frac{10\beta}{\epsilon^2} \cdot \dim_E(\mathcal{F};\epsilon) + 8\log(\delta^{-1}).
\end{aligned}$$

Plugging this back, we obtain

$$\begin{aligned}
\mathrm{Queries}_T^{\mathrm{CB}} \leq &T_\epsilon + \frac{10A^2\beta}{\epsilon^2} \cdot \dim_E(\mathcal{F};\epsilon) + 8A^2\log(\delta^{-1}) \\
\leq &8T_\epsilon^2\beta/A + 6272A^3\beta^3 \frac{\dim_E^2(\mathcal{F},\epsilon)}{\epsilon^2} \cdot \log^2(2/(\delta\epsilon)).
\end{aligned}$$

On the other hand, when the contrary of (16) holds, i.e.,

$$2T_\epsilon + 56A^2\beta \cdot \frac{\dim_E(\mathcal{F}, \epsilon)}{\epsilon} \cdot \log(2/(\delta\epsilon)) \geq \sqrt{AT/\beta}.$$

Squaring both sides and leveraging the inequality $(a + b)^2 \leq 2a^2 + 2b^2$, we obtain

$$8T_\epsilon^2 + 6272A^4\beta^2 \frac{\dim_E^2(\mathcal{F}, \epsilon)}{\epsilon^2} \cdot \log^2(2/(\delta\epsilon)) \geq AT/\beta$$

which leads to

$$T \leq 8T_\epsilon^2\beta/A + 6272A^3\beta^3 \frac{\dim_E^2(\mathcal{F}, \epsilon)}{\epsilon^2} \cdot \log^2(2/(\delta\epsilon)).$$

We note that we always have $\mathrm{Queries}_T^{\mathrm{CB}} \leq T$ and thus

$$\mathrm{Queries}_T^{\mathrm{CB}} \leq T \leq 8T_\epsilon^2\beta/A + 6272A^3\beta^3 \frac{\dim_E^2(\mathcal{F}, \epsilon)}{\epsilon^2} \cdot \log^2(2/(\delta\epsilon)).$$

**Minimizing on $\epsilon$.** Given that the aforementioned proofs hold for any threshold $\epsilon$, we can select the specific value of $\epsilon$ that minimizes the upper bounds. Hence, we deduce the desired result. $\qquad\square$

### A.6. Proof of Theorem 4.2

*Proof of Theorem 4.2.* The upper bound of the number of queries is straightforward: Algorithm 2 is simply running $H$ instances of Algorithm 1, so the total number of queries is simply the sum of these $H$ instances. For bounding the regret, we have

$$
\begin{aligned}
\mathrm{Regret}_T^{\mathrm{IL}} &= \sum_{t=1}^T V_0^{\pi_e}(x_{t,0}) - V_0^{\pi_t}(x_{t,0}) \\
&\leq \sum_{h=0}^{H-1} \sum_{t=1}^T \mathop{\mathbb{E}}_{x_{t,h}, a_{t,h} \sim d_{x_{t,0},h}^{\pi_t}} \left[ Q_h^{\pi_e}(x_{t,h}, \pi_h^{\pi_e}(x_{t,h})) - Q_h^{\pi_e}(x_{t,h}, a_{t,h}) \right] \\
&\leq \sum_{h=0}^{H-1} \sum_{t=1}^T \mathop{\mathbb{E}}_{x_{t,h}, a_{t,h} \sim d_{x_{t,0},h}^{\pi_t}} \left[ Q_h^{\pi_e}(x_{t,h}, \pi_h^+(x_{t,h})) - Q_h^{\pi_e}(x_{t,h}, a_{t,h}) \right] \\
&\quad - \sum_{h=0}^{H-1} \sum_{t=1}^T \mathop{\mathbb{E}}_{x_{t,h} \sim d_{x_{t,0},h}^{\pi_t}} \left[ A_h^{\pi_e}(x_{t,h}, \pi_h^+(x_{t,h})) \right] \\
&\leq H \cdot \mathbb{E}\left[ \mathrm{Regret}_T^{\mathrm{CB}} \right] - \mathrm{Adv}_T.
\end{aligned}
$$

where the first inequality holds by Lemma B.4, and we denote $\pi_h^+(x_{t,h}) = \arg\max_a Q_h^{\pi_e}(x_{t,h}, a)$ in the second inequality. Then, we can plug the upper bound of $\mathrm{Regret}_T^{\mathrm{CB}}$ (Theorem 3.4). Moreover, we need to take a union bound over all $h \in [H]$. $\qquad\square$

## B. Supporting Lemmas

**Lemma B.1** (Kakade & Tewari (2008, Lemma 3)). *Suppose $X_1, \ldots, X_T$ is a martingale difference sequence with $|X_t| \leq b$. Let*

$$\mathrm{Var}_t X_t = \mathrm{Var}(X_t \mid X_1, \ldots, X_{t-1})$$

*Let $V = \sum_{t=1}^T \mathrm{Var}_t X_t$ be the sum of conditional variances of $X_t$ 's. Further, let $\sigma = \sqrt{V}$. Then we have, for any $\delta < 1/e$ and $T \geq 3$,*

$$\Pr\left( \sum_{t=1}^T X_t > \max\{2\sigma, 3b\sqrt{\ln(1/\delta)}\}\sqrt{\ln(1/\delta)} \right) \leq 4\ln(T)\delta.$$

The following lemma is adopted from Foster & Rakhlin (2020, Lemma 3).

**Lemma B.2.** *For any vector $\hat{y} \in [0, 1]^A$, if we define $p$ to be*

$$p(a) = \begin{cases} \frac{1}{A + \gamma\left(\hat{y}(\hat{a}) - \hat{y}(a)\right)} & \text{if } a \neq \hat{a}, \\ 1 - \sum_{a \neq \hat{a}} p(a) & \text{if } a = \hat{a} \end{cases}$$

*where $\hat{a} = \arg\max_a \hat{y}(a)$, then for any $y^\star \in [0, 1]^A$ and $\gamma > 0$, we have*

$$\mathbb{E}_{a \sim p}\left[\left(y^\star(a^\star) - y^\star(a)\right) - \gamma\left(\hat{y}(a) - y^\star(a)\right)^2\right] \leq \frac{A}{\gamma}.$$

**Lemma B.3** (Zhu & Nowak (2022, Lemma 2)). *Let $(Z_t)_{t \leq T}$ to be real-valued sequence of positive random variables adapted to a filtration $\mathfrak{F}_t$. If $|Z_t| \leq B$ almost surely, then with probability at least $1 - \delta$,*

$$\sum_{t=1}^{T} Z_t \leq \frac{3}{2} \sum_{t=1}^{T} \mathbb{E}_t\left[Z_t\right] + 4B \log\left(2\delta^{-1}\right),$$

*and*

$$\sum_{t=1}^{T} \mathbb{E}_t\left[Z_t\right] \leq 2 \sum_{t=1}^{T} Z_t + 8B \log\left(2\delta^{-1}\right).$$

**Lemma B.4** (Performance difference lemma (Agarwal et al., 2019)). *For any two policies $\pi$ and $\pi'$ and any state $x_0 \in \mathcal{X}$, we have*

$$V_0^\pi(x_0) - V_0^{\pi'}(x_0) = \sum_{h=0}^{H-1} \mathbb{E}_{x_h, a_h \sim d_{x_0, h}^\pi}\left[A_h^{\pi'}(x_h, a_h)\right]$$

*where $A_h^\pi(x, a) = Q_h^\pi(x, a) - V_h^\pi(x, a)$ and $d_{x_0, h}^\pi(x, a)$ is the probability of $\pi$ reaching the state-action pair $(x, a)$ at time step $h$ starting from initial state $x_0$.*

**Lemma B.5.** *For any two Bernoulli distributions $\text{Bern}(x)$ and $\text{Bern}(y)$ with $x, y \in [b, 1 - b]$ for some $0 < b \leq 1/2$, the KL divergence of them is upper bounded as follows:*

$$\text{KL}\left(\text{Bern}(x), \text{Bern}(y)\right) \leq \frac{2(x - y)^2}{b}.$$

*Proof of Lemma B.5.* Denote $\Delta = x - y$. Then, by definition, we have

$$\begin{aligned} \text{KL}\left(\text{Bern}(x), \text{Bern}(y)\right) =& x \ln \frac{x}{y} + (1 - x) \ln \frac{1 - x}{1 - y} \\ =& x \ln \frac{x}{x - \Delta} + (1 - x) \ln \frac{1 - x}{1 - x + \Delta} \\ =& x \ln\left(1 + \frac{\Delta}{x - \Delta}\right) + (1 - x) \ln\left(1 - \frac{\Delta}{1 - x + \Delta}\right) \end{aligned}$$

Since $\ln(1 + x) \leq x$ for all $x > -1$, we have

$$\begin{aligned} \text{KL}\left(\text{Bern}(x), \text{Bern}(y)\right) \leq& x \cdot \frac{\Delta}{x - \Delta} - (1 - x) \cdot \frac{\Delta}{1 - x + \Delta} \\ =& \Delta \cdot \left(\frac{x}{x - \Delta} - \frac{1 - x}{1 - x + \Delta}\right) \\ =& \Delta \cdot \left(\frac{\Delta}{x - \Delta} + \frac{\Delta}{1 - x + \Delta}\right) \\ \leq& \Delta^2 \cdot \left(\frac{1}{y} + \frac{1}{1 - y}\right) \\ \leq& \frac{2\Delta^2}{b}. \end{aligned}$$

$\square$

