# OpenReview forum: "Contextual Bandits and Imitation Learning with Preference-Based Active Queries"
_ICML.cc/2023/Workshop/ILHF — ILHF Workshop ICML 2023_

### Official Review · Reviewer_v4M7 · 2023-06-09

**Rating:** 7
**Confidence:** 2

**Review:**

### Quality & Clarity

The paper is well-written and easy to understand – I didn't see any typos, and it was structured well. The motivation is clear.

I did not verify the correctness of the proofs but the results seem to make sense intuitively.

### Significance & Originality

I'm not very familiar with the theoretical side of this area – because of this, I'm not able to meaningfully assess the significance or originality of this work in the current research landscape. That being said, to my knowledge, the results are novel and seem like a meaningful contribution.

### Related work

Reading this paper, various papers from the more applied side came to mind. You can use your own judgement as to which ones you feel like are relevant enough to reference!

> In other areas, such as robotics, learning from human feedback is also not easy. (Ross et al., 2013; Laskey et al., 2016) pointed out that querying human feedback in the learning loop is challenging, and extensively querying for feedback puts too much burden on the human experts.

Christiano et al., "Deep reinforcement learning from human preferences" (2017) does seem to successfully use reasonable amounts of human data to train complex robotic behaviors. More recently, Zhang et al. "Time-Efficient Reward Learning via Visually Assisted Cluster Ranking" (2022), also proposes an approach for reducing the time burden on human experts in learning from human feedback information for robotics settings that seems promising.

In terms of being able to do better than suboptimal demonstrations without reward information, you might want to consider Brown et. al., "Extrapolating Beyond Suboptimal Demonstrations via Inverse Reinforcement Learning from Observations" (2019), Brown et. al., "Better-than-Demonstrator Imitation Learning via Automatically-Ranked Demonstrations" (2019), and Brown et. al. "Safe Imitation Learning via Fast Bayesian Reward Inference from Preferences" (2020).

An additional work that may be relevant to the discussions about different kinds of human feedback is Jeon et. al., "Reward-rational (implicit) choice: A unifying formalism for reward learning" (2020). They propose a framework that categorizes and unifies many human feedback types (but may not explicitly cover action feedback).

---

### Official Review · Reviewer_xsdi · 2023-06-18
**Theoretical Paper, Weak Motivation, No Empirical Results**

**Rating:** 6
**Confidence:** 2

**Review:**

This paper proposes a preference-based active querying methods for contextual bandit problems and for imitation learning. The paper has a great deal of theoretical content, which I didn't verify carefully, but I also did not spot any errors.

I have three major concerns about the paper:

1) The paper should motivate the problem better. What are some cases where the agent needs to either ask a question or select two options (and not observe the outcome)? Imitation learning setup seems more practical, but this same comment could apply there too.
2) The paper has some theoretical results, but no experiments or any empirical results. It would be nice to validate the theory with experiments.
3) The feedback adopted in this paper is not really implicit. On the contrary, it is rather explicit. However the workshop is about implicit feedback (that being said, I acknowledge the fact that the invited speakers of the workshop include people who have worked on preference-based learning).

The paper could also cite the following papers that seem relevant:
Cohn et al.'s "Comparing action-query strategies in semi-autonomous agents" and Myers et al.'s "Active Reward Learning from Online Preferences" are relevant due to the interactive setup in which the agent has to choose between asking a question and taking an action, and the goal is to ask as few questions as possible while still attaining good performance.
Brown et al.'s "Better-than-Demonstrator Imitation Learning via Automatically-Ranked Demonstrations" may be relevant as it is an(other) example of how preference-based learning can outperform the suboptimal annotators.

---

### Decision · Program_Chairs · 2023-06-20

Accept